# Heart enhancers with deeply conserved regulatory activity are established early in zebrafish development

Xuefei Yuan[1,2,3], Mengyi Song[1,2,3], Patrick Devine[4,5], Benoit G. Bruneau[4,6], Ian C. Scott[2,3] & Michael D. Wilson [1,3]

During the phylotypic period, embryos from different genera show similar gene expression patterns, implying common regulatory mechanisms. Here we set out to identify enhancers involved in the initial events of cardiogenesis, which occurs during the phylotypic period. We isolate early cardiac progenitor cells from zebrafish embryos and characterize 3838 open chromatin regions specific to this cell population. Of these regions, 162 overlap with conserved non-coding elements (CNEs) that also map to open chromatin regions in human. Most of the zebrafish conserved open chromatin elements tested drive gene expression in the developing heart. Despite modest sequence identity, human orthologous open chromatin regions recapitulate the spatial temporal expression patterns of the zebrafish sequence, potentially providing a basis for phylotypic gene expression patterns. Genome-wide, we discover 5598 zebrafish-human conserved open chromatin regions, suggesting that a diverse repertoire of ancient enhancers is established prior to organogenesis and the phylotypic period.

[1] Program in Genetics and Genome Biology, The Hospital for Sick Children, Toronto, ON M5G 0A4, Canada. [2] Program in Developmental and Stem Cell Biology, The Hospital for Sick Children, Toronto, ON M5G 0A4, Canada. [3] Department of Molecular Genetics, University of Toronto, Toronto, ON M5S 1A8, Canada. [4] Gladstone Institutes, San Francisco, CA 94158, USA. [5] Department of Pathology, University of California, San Francisco, San Francisco, CA 94143-0511, USA. [6] Department of Pediatrics and Cardiovascular Research Institute, University of California, San Francisco, San Francisco, CA 94158, USA. Correspondence and requests for materials should be addressed to I.C.S. (email: ian.scott@sickkids.ca) or to M.D.W. (email: michael.wilson@sickkids.ca)

The developmental hourglass model predicts a phylotypic stage during mid-embryogenesis when species within the same phylum display the greatest level of morphological similarities[1,2]. The hourglass model is also supported by comparative transcriptomic studies that demonstrated that the most conserved gene expression patterns occur at the phylotypic stage[3–5]. The idea that conserved phylotypic gene expression is established through conserved enhancers is supported by several comparative epigenomic studies[6–9]. While most molecular studies of the phylotypic period have focused on whole embryos, recent evidence suggests that the exact developmental timing of maximal conservation varies in a tissue-specific manner[8]. We are only beginning to understand how conserved transcriptional programs for individual developmental lineages are set up prior to the phylotypic stage.

The heart, derived from the cardiac mesoderm, is the first organ formed during embryogenesis. Heart development is orchestrated by conserved cardiac transcription factors (TFs) binding to cis-regulatory elements (CREs)[10,11]. Crucial cardiac specification events occur during early embryogenesis[12–15]. For example, distinct subtypes of mouse cardiac progenitors emerge within the gastrula stage preceding the expression of the canonical cardiac progenitor marker Nkx2.5, long before any organ structure is formed[12–14]. However, how this potential early cardiac specification is controlled by enhancer elements, and the extent to which this process is evolutionarily conserved, is not known.

Heart enhancers have been extensively characterized in studies utilizing genome-wide profiling techniques including chromatin immunoprecipitation followed by DNA sequencing (ChIP-seq)[16–21], computational predictions[22,23], and in vivo validation in mouse and zebrafish embryos[16,17,19,22,23]. To date, the majority of in vivo heart enhancer discoveries and validation experiments were performed in embryos following establishment of the heart chamber structures or in adult hearts[16,17]. In vitro differentiation of embryonic stem cells (ESCs) into cardiac progenitors have also yielded insights into the early cardiac development and have enabled the discovery of cardiac enhancers[18,20,21]. However, more work is needed to identify the CREs that regulate the early differentiation of mesoderm progenitors to cardiac lineages in the context of the developing embryo.

Despite the highly conserved cardiac TFs necessary for heart development, heart enhancers identified at mouse E11.5 show limited phylogenetic conservation compared to brain enhancers identified at the same developmental stage[8,16]. However, analysis of putative enhancers in mesoderm cells, derived from embryonic stem cells, show higher evolutionary constraint than the enhancers identified after organogenesis[8]. This suggests that the regulatory elements that establish the conserved cardiac transcriptional program may exist at the initial stages of heart development, presumptively during the time window from naive mesoderm to cardiac progenitors.

Here, we set out to discover enhancers that are active in cardiac progenitor cells prior to the expression of the cardiac progenitor marker Nkx2.5. We generate a zebrafish GFP reporter line driven by a mouse Smarcd3 enhancer (Smarcd3-F6) that is active in early gastrulating mouse mesoderm[12], in order to enrich for early zebrafish cardiac progenitors. We profile gene expression and the accessible chromatin landscape of GFP positive cells using single cell mRNA-seq, the assay for transposase-accessible chromatin using sequencing (ATAC-seq), lineage tracing, and RNA in-situ hybridization, both in wild-type embryos and following knockdown of the essential cardiac transcription factors Gata5/6. Results from these experiments indicate that we have purified a population of cells enriched for cardiac progenitors. Using direct and indirect DNA alignments[24,25], we identify accessible chromatin regions shared between zebrafish and human. We find that these conserved accessible chromatin elements were highly associated with developmental transcription factors that are regulated by polycomb repressive complex 2 (PRC2). We confirm the cardiac activity and functional conservation of many anciently conserved open chromatin regions using in vivo reporter assays. In sum, our study identifies a set of conserved cardiac enhancers established prior to the phylotypic period, before the heart and other organ primordia appear, potentially providing a basis for common gene expression patterns across genera. Furthermore, we uncover ~ 6000 anciently conserved open chromatin regions that likely serve as enhancers for other cell lineages.

## Results

**Isolation of zebrafish cardiac progenitor cells.** To examine the earliest events that contribute to cardiac lineage specification in zebrafish, we needed a means to isolate early cardiac progenitor cells in vivo. To identify a zebrafish marker that could drive GFP expression in cardiac progenitor cells prior to nkx2.5 expression, we tested a recently described early mouse cardiac enhancer, Smarcd3-F6[12] in our zebrafish model. Lineage tracing experiments demonstrated that this enhancer labeled cardiac progenitor cells prior to Nkx2.5 expression in mouse embryos[12]. We also found that the Smarcd3-F6 region was enriched for the active enhancer mark H3K27ac and contains a CRE co-bound by several conserved cardiac TFs (GATA4, NKX2.5, TBX5) in mouse ESC differentiated cardiac precursors (CP) and cardiomyocytes (CM)[18,21] (Supplementary Fig. 1a).

To test if the Smarcd3-F6 enhancer functions as an early marker of cardiac progenitors in zebrafish, we generated a Tg (Smarcd3-F6:EGFP) transgenic line (Fig. 1a). Due to the lag time between transcription and GFP accumulation, we conducted RNA in-situ hybridization against gfp in order to detect enhancer activity at early developmental times. We found that gfp signal could be detected as early as 6 h post-fertilization (hpf) along the embryonic margin (Fig. 1b), which contains mesendodermal progenitors including future cardiac cells[26]. Over the course of gastrulation, GFP positive cells migrated to encompass positions in the anterior and posterior lateral plate mesoderm (ALPM and PLPM) (Fig. 1b, c). Co-immunostaining comparing Tg(Smarcd3-F6:EGFP) and Tg(nkx2.5:ZsYellow) expression indicated that the Smarcd3-F6 enhancer marked almost all cardiac mesoderm expressing nkx2.5 at early somite stages (13 hpf) (Fig. 1d). Tg(Smarcd3-F6:CreERT2) lines were generated to trace the fate of Smarcd3-F6 labeled cells. By crossing Tg(Smarcd3-F6:CreERT2) to a Tg(βactin2-loxP-dsRed-loxP-GFP) reporter line, we found that following 4-hydroxytamoxifen (4-HT) addition at 8 hpf, cells labeled by the Smarcd3-F6 enhancer contributed to heart formation (Supplementary Fig. 1). Although the Smarcd3-F6 enhancer shows limited mammalian, and no zebrafish sequence conservation (Supplementary Fig. 1a), the early labeling of cardiac lineages in zebrafish indicated that this enhancer would allow us to isolate a cell population enriched for cardiac progenitors.

To further characterize the population marked by the Smarcd3-F6 enhancer, we conducted bulk mRNA-seq and ATAC-seq at 10 hpf on Smarcd3-F6 labeled (GFP+) and unlabeled (GFP−) cells (Fig. 1e). mRNA-seq revealed 316 genes differentially expressed between the GFP+ and GFP− population (FDR < 0.05, absolute log2FC > 1). Known cardiac (gata5, nkx2.7, tbx20, hand2) and endoderm (sox32, sox17) markers showed significantly higher expression in Smarcd3-F6 labeled cells while ectoderm and axial mesoderm genes were relatively depleted (Fig. 1f). Consistent with these results, genes showing higher expression in the GFP+ population were enriched for processes related to cardiovascular and endoderm development, whereas genes enriched in

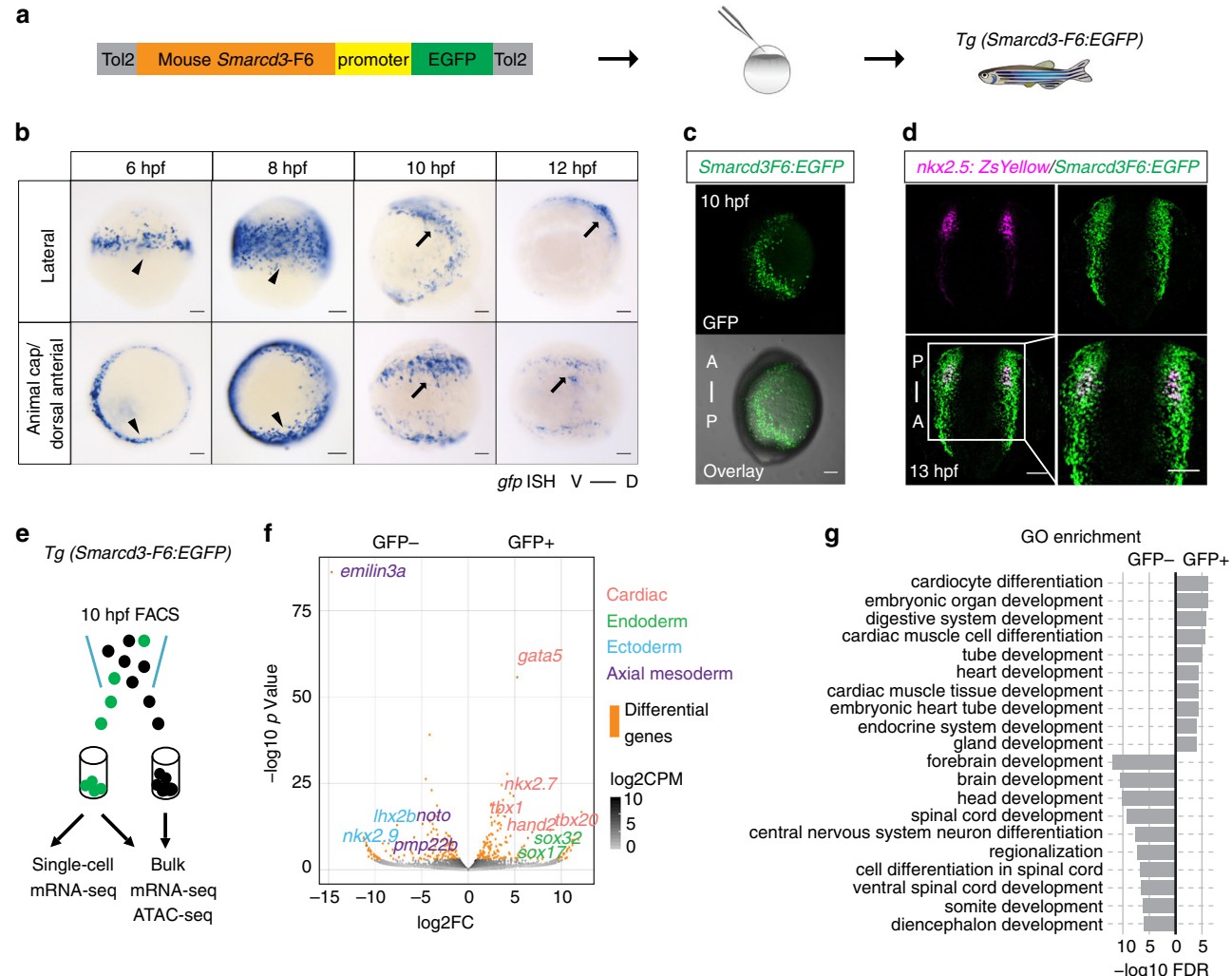

**Fig. 1** Mouse *Smarcd3*-F6 enhancer labels early cardiac progenitors in zebrafish. **a** Generation of *Tg(Smarcd3-F6:EGFP)* zebrafish line **b** In-situ hybridization against *gfp* transcripts on *Tg(Smarcd3-F6:EGFP)* transgenic embryos. *Smarcd3-F6* enhancer marks lateral margins (arrowheads) during gastrulation and ALPM regions (arrows) after gastrulation. **c** Native GFP expression in *Tg(Smarcd3-F6:EGFP)* embryos at 10 hpf. Embryos are shown in lateral views. **d** Immunostaining of GFP and ZsYellow on *Tg(Smarcd3-F6:EGFP)* and *Tg(nkx2.5:ZsYellow)* double transgenic embryos. Cells expressing ZsYellow were marked by GFP as well. **e** Workflow of mRNA-seq and ATAC-seq experiments. **f** Volcano plot showing genes differentially expressed between *Smarcd3-F6*: GFP+ and *Smarcd3-F6*:GFP- populations (FDR < 0.05, absolute log2FC > 1). **g** Top 10 most enriched GO terms obtained from genes that were significantly more highly expressed in *Smarcd3-F6*:GFP+ or *Smarcd3-F6*:GFP- populations. All scale bars represent 100 μm

GFP− cells were enriched for those involved in nervous system development (Fig. 1g).

To further dissect the putative cardiac progenitors within the 10 hpf *Smarcd3*-F6 labeled population, we performed single-cell mRNA-seq on 96 GFP+ cells. The average of all single-cell transcriptomes correlated well with the bulk mRNA-seq results (Pearson correlation $R = 0.93$, $P < 2.2e−16$) (Supplementary Fig. 2a). Unsupervised clustering using genes differentially expressed between GFP+ and GFP− populations grouped cells into three broad clusters, which represented putative endodermal (*gata5*+, *sox17*+, *sox32*+), mesodermal (*gata5*+, *sox17*−, *sox32*−) and ectodermal (*gata5*−, *sox2*+, *sox3*+) populations (Supplementary Fig. 2b). Within the mesoderm cluster, we could identify a potential cardiac subgroup co-expressing the known cardiac genes *gata5, hand2, tbx20,* and *tmem88a* (Supplementary Fig. 2b, c). Together, our transcriptome analyses demonstrated that cells labeled by the *Smarcd3*-F6 enhancer are enriched for cardiac lineages, with a putative cardiac progenitor population apparent by as early as 10 hpf.

**ATAC-seq analysis of zebrafish cardiac progenitor cells**. Open chromatin profiles often identify the genomic regions where TFs and their co-factors bind and function[27]. Using ATAC-seq, we detected 155,879 open chromatin regions (ATAC-seq peaks) in the GFP+ population and 153,777 in the GFP− population. After conducting differential analysis, we found most ATAC-seq peak regions ($n = 195,466$) shared similar ATAC-seq signals in both GFP+ and GFP− populations. 5471 peaks showed significant quantitative differences (Fig. 2a) with 3838 peaks specifically increased in *Smarcd3*-F6 labeled cells ('GFP+ specific'), and 1633 peaks specifically increased in unlabeled cells ('GFP− specific') (Supplementary Data 1).

Our ATAC-seq peaks significantly overlap with active chromatin marks found at promoters (H3K4me3, $P = 0.001$, permutation test by GAT[28]) and enhancers (H3K27ac, $P = 0.001$; H3K4me1, $P = 0.001$, permutation test by GAT[28]) that were previously identified from ChIP-seq experiments on whole zebrafish embryos of similar stages[6]. We found that 69% of accessible zebrafish regions did not overlap with active chromatin

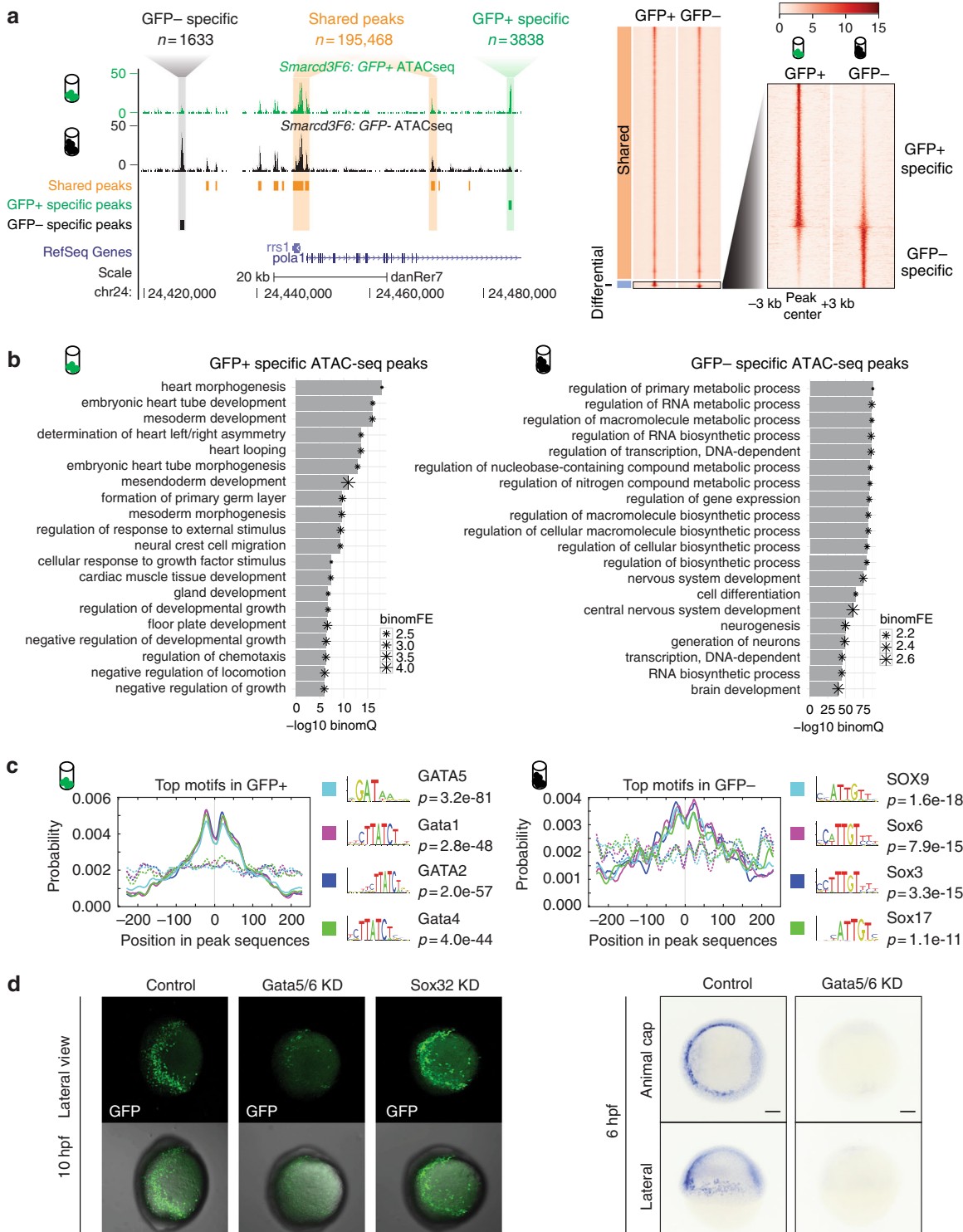

**Fig. 2** Open chromatin landscape of *Smarcd3*-F6 labeled population. **a** Genome browser view and heatmap showing ATAC-seq peaks shared between GFP+ and GFP− populations or enriched in one population. In the heatmap, read intensity for regions within 3 kb of the peak center was plotted for each peak. **b** Barplot showing the 20 most enriched GO biological function terms obtained from GFP+/− specific peaks using GREAT analysis. **c** Probabilities of the top 4 enriched motifs within the GFP+/− specific peaks calculated by CentriMo[76]. Each curve shows the probability of the best match to a given motif occurring at a given position in the input sequences. Solid lines represent probabilities calculated from query peak sets (GFP+/− specific peaks) while dash lines show that from the background sequences (shared peaks). **d** (Left) Confocal images of native GFP expression in control (uninjected), Gata5/6 KD (injected with Gata5/6 morpholinos) and Sox32 KD (injected with CAS morpholino) *Tg(Smarcd3-F6: EGFP)* transgenic embryos at 10 hpf; (Right) *gfp in-situ* on *Tg(Smarcd3-F6: EGFP)* embryos of 6 hpf that were uninjected (control) or injected with Gata5/6 morpholinos. All staining and imaging were performed under the same condition for the control and KD groups. All scale bars represent 100 μm

marks (Supplementary Fig. 3a). The prevalence of open chromatin regions with low levels of active chromatin marks has been readily observed in human cell lines and associated with a signature of motif-dependent binding characteristic of repressive chromatin states[29]. It is likely that the lack of overlap of our accessible chromatin regions with active chromatin marks obtained from whole embryos will involve such regions in addition to being due to technical reasons such as having purified a small subset of cells away from the bulk embryo prior to ATAC-seq.

Overall both the GFP+ specific and GFP− specific open chromatin regions were depleted for proximal promoter (GFP+: adjusted $P = 1.83e–21$, GFP-: adjusted $P = 6.70e–15$, Fisher's exact test, two-sided) and exonic regions (GFP+: adjusted $P = 3.51e–17$, GFP−: adjusted $P = 4.81e–11$, Fisher's exact test, two-sided) and enriched for introns (GFP+: adjusted $P = 7.35e–12$, GFP−: adjusted $P = 1.00e–10$, Fisher's exact test, two-sided) as compared to the genomic distribution of all ATAC-seq peaks (Supplementary Fig. 3b). Using the enrichment tool GREAT[30], we found that the GFP+ specific open chromatin regions were enriched for heart development-related processes, based on proximity to genes (heart morphogenesis: FDR = 1.09e–18, embryonic heart tube development: FDR = 8.91e–17, binomial test, Fig. 2b). GFP− specific open chromatin regions were enriched for metabolic, gene expression and neural development-related processes (regulation of primary metabolic process: FDR = 5.60e–89, regulation of transcription, DNA-dependent: FDR = 1.40e–86, nervous system development: FDR = 1.35e–75, binomial test, Fig. 2b). Overall, the transcriptional profiles and open chromatin regions enriched in the GFP+ and GFP− specific populations further indicate that the *Smarcd3*-F6 enhancer marks cardiac progenitor cells.

We then asked which TF motifs were overrepresented in GFP+ specific peaks. We found that GATA motifs showed the strongest enrichment, consistent with the crucial roles GATA factors play in heart and endoderm development[31–34] (Fig. 2c). In contrast, GFP- specific peaks were most highly enriched for SOX motifs (Fig. 2c). Like *Gata4* and *Gata6* in mice[33,34], *gata5* and *gata6* play redundant but critical roles in zebrafish heart formation[31,32]. To test if the activity of the *Smarcd3*-F6 enhancer is regulated by Gata5 and Gata6 in zebrafish, we performed *gata5* and *gata6* knock-downs by injecting previously validated morpholinos[32] into *Tg(Smarcd3-F6: EGFP)* embryos. Supporting our motif enrichments, we found that *Smarcd3*-F6 enhancer activity requires Gata5 and Gata6 function (Fig. 2d). In contrast, knocking down an early active SOX transcription factor (Sox32)[35] did not ablate the GFP signal (Fig. 2d).

**Comparative analyses reveal conserved accessible chromatin.** To identify regions of open chromatin that are conserved between zebrafish and human, we used two well-defined conserved non-coding element (CNE) datasets, zCNE[24] and garCNE[25]. Both of these two resources contain conserved regions identified using direct alignment and indirect homology bridged by intermediate species (Fig. 3a and Supplementary Table 1). We associated zebrafish ATAC-seq peaks to CNEs if they overlapped a zebrafish-human or zebrafish-mouse CNE. Most accessible chromatin-associated CNEs (~ 70%) were fully contained within ATAC-seq peaks. On average 30% of the length of these ATAC-seq peaks were comprised of CNEs. Within a total of 200,937 ATAC-seq peaks, we found 6294 (3.1%) or 6047 (3.0%) shared sequence conservation with human or mouse respectively (see Methods, Supplementary Fig. 3c and Supplementary Table 2 for details). Of these 6294 zebrafish-human ATAC-seq peaks, 176 were GFP+ specific, and 264 were GFP− specific.

We found that GFP− specific peaks were ~ 4 times more enriched for conserved regions than the GFP+ ones ($P = 1.40e–46$ using human CNEs; Chi-square test, two-sided) (Fig. 3a). Previous work has shown that 30–45% of forebrain, midbrain and limb enhancers overlapped regions of extremely high sequence constraint, in contrast to only ~ 6% of cardiac enhancers[16]. Given that GFP+ and GFP− populations were enriched for cardiac and brain lineages respectively (Fig. 1f), our observations were consistent with previous finding[8,16].

Comparing genomic features (e.g., TF binding, chromatin accessibility, histone modifications) between species is a potentially powerful way to identify conserved regulatory function within alignable sequences for specific tissues or cell types[36–41]. For ATAC-seq peaks overlapping CNEs, we asked if the orthologous human or mouse CNEs also contained accessible chromatin according to DNase I hypersensitivity sites (DHS) reported by ENCODE3[42] (Fig. 3b). We refer to these conserved accessible chromatin regions connected through CNEs as aCNEs. We found that the majority of the zebrafish ATAC-seq peaks overlapping zebrafish-human or zebrafish-mouse CNEs were classified as aCNEs (89% for human (5598/6294) and 79% (4747/6047) for mouse; Fig. 3b, c). Overall ~ 3% of the total zebrafish ATAC-seq peaks were identified as aCNEs that are shared between zebrafish and human, or zebrafish and mouse. On average, one zebrafish aCNE corresponded to 1.6 human and or 1.4 mouse aCNEs (Supplementary Fig. 3c, see Methods for details)

As aCNEs were derived from CNE datasets, in which all coding sequences had been carefully excluded[24,25], we found that aCNEs were depleted for exonic regions (0.78% versus 4.7%, adjusted $P = 3.95e–50$, Fisher's exact test, two-sided) and were enriched for intronic regions (25.0% versus 21.7%, adjusted $P = 5.15e–07$, Fisher's exact test, two-sided) compared to all ATAC-seq peaks. We did not see an enrichment for promoter regions in aCNEs compared to all ATAC-seq peaks (9.34% versus 9.42%, adjusted $P = 1$, Fisher's exact test, two-sided). While ATAC-seq peaks at promoters were more conserved than those not at promoters ($P = 7.82e–289$, Wilcoxon test, two-sided), promoter aCNEs did not show higher sequence constraint than non-promoter aCNEs ($P = 0.19$, Wilcoxon test, two-sided) (Supplementary Fig. 3d).

We asked whether any genomic features could distinguish aCNEs from bulk open chromatin regions. First, we found that conserved ATAC-seq peaks had a wider boundary (average 706 bp compared to 475 bp, $P = 0.001$; permutation test), stronger open chromatin signals ($P = 2.84e–271$; Wilcoxon test, two-sided) and slightly higher GC content (average 46% compared to 44%, $P = 0.001$; permutation test) than the bulk zebrafish ATAC-seq peaks (Supplementary Fig. 3e, f). When we separated promoter and non-promoter aCNEs for analysis, we observed similar results regarding interval length, GC content and open chromatin signal intensity (Supplementary Fig. 3e, f).

We compared our aCNEs to ultraconserved non-coding elements in the human genome[43], which were defined as sequence elements with no mismatches for at least 200 base pairs between orthologous regions in the human, rat and mouse genomes. While 30% of ultraconserved elements overlap human aCNEs, only 3% of the aCNEs overlap ultraconserved elements (Supplementary Data 2). The fact that nearly 40% (2228/5598 for zebrafish-human aCNEs) of the aCNEs were found using indirect alignments indicates that many of these ancient aCNEs show limited sequence conservation and would likely have been overlooked using standard multiple genome alignment methods.

**aCNEs drive expression in the developing heart.** To gain insights into the cardiac enhancer activity of aCNEs, we

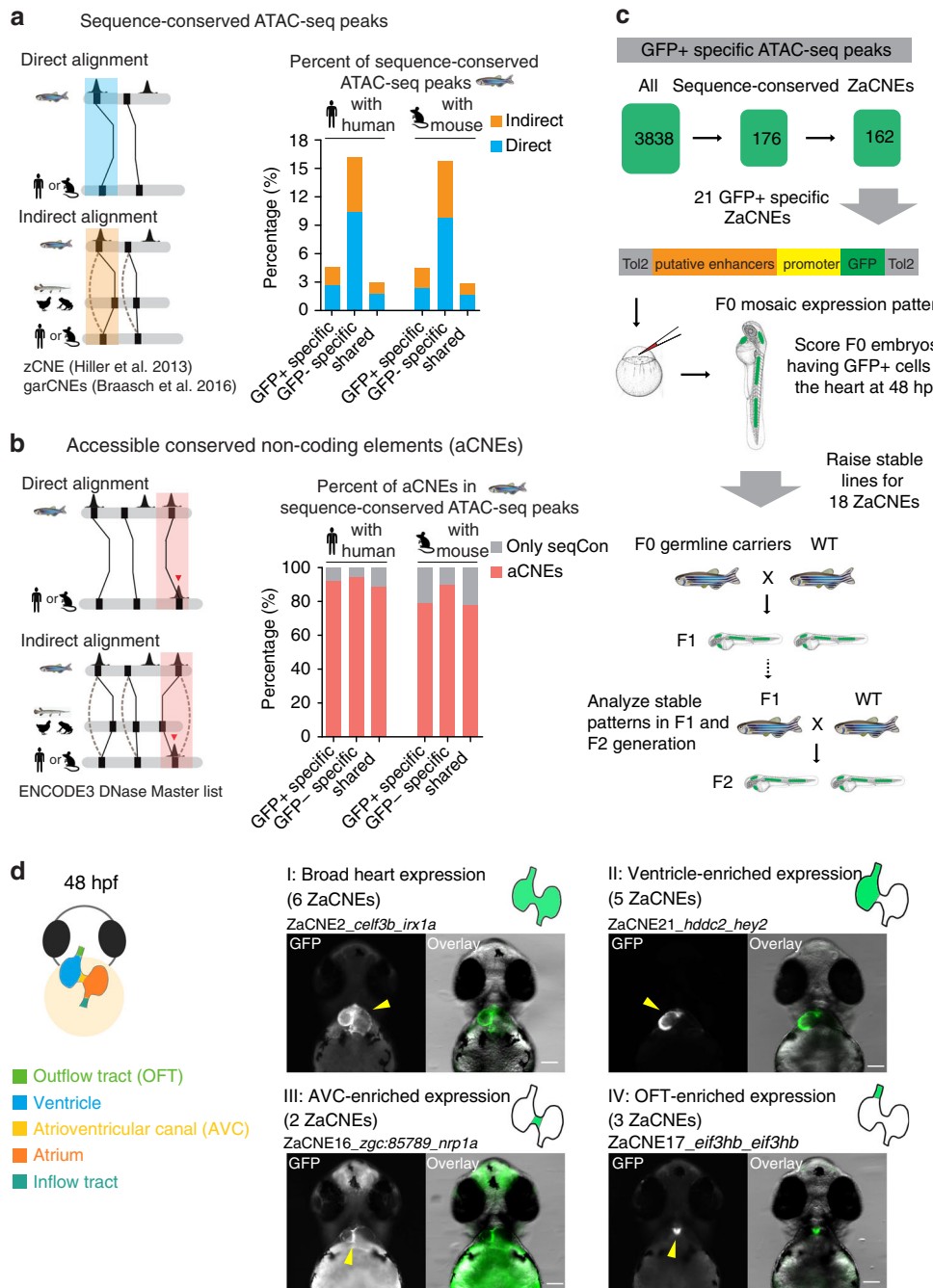

**Fig. 3** Identification and in vivo characterization of accessible conserved non-coding elements (aCNEs). **a** Barplot showing the percentage of conserved regions in all GFP+/− specific and shared ATAC-seq categories. Solid lines in the cartoon indicate direct sequence alignment and dotted lines showcases of indirect alignment bridged by intermediate species or ancestry reconstruction. **b** Barplot showing the aCNEs/sequenced conserved (seqCon) percentage. Most seqCon open chromatin regions were identified as having shared orthologous accessible chromatin. **c** Strategy for identifying and characterizing GFP+ specific aCNEs in zebrafish. Stable transgenic lines for 18 aCNEs were characterized. **d** Cartoon showing the zebrafish heart anatomy at 48 hpf (left panel). Heart expression patterns observed in ZaCNEs transgenic lines were classified into 4 categories (right panel), including I: broad expression in the whole heart; II: ventricle-enriched expression; III: atrioventricular (AVC)-enriched expression; IV: outflow tract (OFT)-enriched expression. One representative example of each category was shown and the numbers of ZaCNEs fall in each category are indicated in the brackets. The two closest genes near each ZaCNEs are indicated. All images were taken from the ventral view and all scale bars represent 100 μm

compared the 281 GFP+ human aCNEs to putative heart enhancers predicated using curated epigenomic data[22]. We found a significant overlap between them (78/281, $P = 1.68e{-}3$, Fisher's exact test, two-sided) when using all human DNase I sites as background. We next examined elements tested in the VISTA enhancer database[44], which is the most comprehensive database of functionally validated enhancers (2017/07/29 release;

https://enhancer.lbl.gov/). When we took a global view of all aCNEs from the human perspective, we found roughly 11% (958/8866) of human aCNEs were included in the VISTA Enhancer Browser. While less than 35% of the human regions (953/2787) profiled by VISTA were functionally validated as positive enhancers, two thirds of human aCNEs (640/958) tested were reported as positive enhancers, indicating a significant

enrichment of active developmental enhancers amongst aCNEs ($P = 9.02e{-}26$, Fisher's exact test, two-sided).

To determine if some conserved open chromatin elements are bound by cardiac TFs during heart development, we collected published ChIP-seq and ChIP-exo data for GATA4, NKX2.5, TBX5, HAND2, MEF2A, and SRF conducted in mouse embryonic hearts or cardiac cell types[19,21,45,46] (Supplementary Data 3). We found that among the 134 GFP+ specific ATAC-seq peaks that were aCNEs shared with mouse, 49 (37%) are bound by at least one cardiac TF at the orthologous regions of mouse genome and 34 (25%) are CREs that are bound by more than one cardiac TF (Supplementary Data 2). In contrast, cardiac TF binding was rarely observed in GFP− specific open chromatin regions (Supplementary Fig. 3f and Supplementary Data 2).

To assess the in vivo function of the GFP+ specific aCNEs, we used a transgenic reporter assay to test their activity during zebrafish embryonic development (Fig. 3c). The regions we selected included both direct (13 regions) and indirect (8 regions) alignments between zebrafish and human (Supplementary Data 4). We cloned the 21 zebrafish regions into a Tol2-based GFP enhancer detection vector and injected the constructs into zebrafish embryos (Fig. 3c). We found that 18/21 regions drove heart expression in at least 30% of the F0 embryos injected (Supplementary Fig. 4), suggesting that they are active enhancers in embryonic heart development. Within this set of 18 enhancers, 11 zebrafish aCNEs (ZaCNEs) were located near known cardiac genes, nine of which overlapped the experimentally determined binding sites (ChIP-seq peaks) of one or more cardiac TFs (GATA4, NKX2.5, TBX5, HAND2, MEF2A, and SRF) in mouse hearts or cardiac cell types[19,21,45,46] (Supplementary Data 4). Four of the seven selected ZaCNEs with no known cardiac gene association had experimentally determined binding sites (ChIP-seq peaks) of one or more cardiac TF (Supplementary Data 4).

We raised stable transgenic lines for the 18 ZaCNEs that passed the 30% threshold in the F0 assays, to verify their cardiac activity (Fig. 3c). We obtained multiple independent alleles for all ZaCNEs except for ZaCNE18 for which only one transgene germline carrier was identified (Supplementary Data 4). Despite the random transgene integration mediated by the Tol2 transposon system, we observed consistent GFP expression in hearts in multiple alleles of the same enhancer lines for 15/17 ZaCNEs, with ZaCNE4 and ZaCNE7 being the exceptions (Supplementary Fig. 5). We also assayed the presumptive phylotypic period (24 hpf) and found that 16/18 drove expression in the linear heart tube (Supplementary Fig. 5e). These results from stable transgenic embryos demonstrated high accordance with those obtained from the F0 generation.

We classified the heart expression observed in the ZaCNE transgenic lines into 4 major categories (Fig. 3d and Supplementary Fig. 5a). Some heart enhancers broadly labeled all heart structures (Category I), while others showed specific or enhanced expression in the ventricle (category II), atrioventricular canal (category III) or outflow tract (category IV) (Fig. 3d and Supplementary Fig. 5). These results confirmed the diverse expression driven by aCNEs and suggested that they may play distinct roles in regulating heart gene expression.

We asked if certain motifs were enriched in each enhancer category. With the exception of category III (atrioventricular canal), which only contains two sequences, enriched motifs were identified in all other categories. GATA, SMAD, RAR/RXR, ZNF263, and RREB1 motifs were found to be enriched in more than one enhancer category, suggesting that they may represent shared features of early cardiac enhancers (Supplementary Fig. 5b). Category IV (outflow tract enhancers) had the most distinct motif signature, with FOX (FOXO1, FOXP1, FOXK1 etc.) and homeobox (HOXA2, EMX2, PDX1, etc.) motifs showing strong enrichment (Supplementary Fig. 5b). Interestingly, Foxp1 has been shown to be required for outflow tract morphogenesis[47].

**aCNEs share conserved early cardiac activities**. We next aimed to determine if human-zebrafish orthologous aCNEs were functionally conserved with respect to their ability to control spatial-temporal gene expression patterns. We chose four aCNEs near the essential cardiac genes *hand2/HAND2* (aCNE1), *tbx20/TBX20* (aCNE20) and *mef2cb/MEF2C* (aCNE5, aCNE19). All of these zebrafish sequences drove robust and specific heart expression in stable transgenic lines (ZaCNE1, ZaCNE5, ZaCNE19, ZaCNE20) (Fig. 4a, b and Supplementary Fig. 6a, b).

When tested in vivo, the four orthologous human aCNE (HaCNE) sequences also drove GFP expression in the hearts of zebrafish transgenic lines, with 3 of them (HaCNE1, HaCNE5, HaCNE20) demonstrating activities similar to that of their zebrafish orthologs (Fig. 4a, b and Supplementary Fig. 6a). At 48 hpf, the GFP expression in the four human aCNE transgenic lines was seen in both chambers and would be classified as pan-cardiac enhancers (category I), but we noticed that the human sequences tended to drive weaker or more mosaic expression than their zebrafish orthologous sequences (Fig. 4a, b and Supplementary Fig. 6a, b).

Due to the long perdurance of GFP protein, we next used *gfp* RNA in-situ to characterize the spatiotemporal dynamics of the aCNE activity. We found that both aCNE1 and aCNE20 pairs were active in cardiac lineages at early somite stages (13 hpf, prior to formation of the linear heart tube), 24 hpf, as well as 48 hpf (Fig. 4c, d). Despite marking a broad population within the ALPM (cardiac domain) at 13 hpf, we found that orthologous pairs of enhancers labeled anatomical subregions of the heart in a similar manner at 48 hpf (Fig. 4c, d). The two ZaCNE1 and HaCNE1 enhancers both labeled ventricles and the inner curvatures of atria (Fig. 4c). For the aCNE20 orthologs, the strongest activity of both was seen in the inner curvature of ventricles and atrioventricular canal regions (Fig. 4d).

Our motif enrichment analyses suggested that GATA factors act as important regulators in the GFP+ population. Within GFP+ specific zebrafish aCNEs conserved with human, 66% (107/162) have at least one GATA motif and ~ 30% (52/162) have more than one GATA motif (Supplementary Data 5). In contrast, while using the same threshold, no significant GATA motifs were found in the conserved GFP− specific zebrafish aCNEs (Supplementary Data 5). To test the functional importance of the GATA motif in aCNEs, we mutated an aligned GATA motif within zebrafish and human aCNE1 regions (near the *hand2/ HAND2* locus) and compared their activities with that of the wild-type sequences (Supplementary Fig. 6c). In F1 stable transgenic lines, we found that both zebrafish and human aCNE1 sequences with mutated GATA motif drove much weaker GFP expression in zebrafish hearts compared to the wild-type sequences (Supplementary Fig. 6c).

These results together demonstrate that despite sequence divergence between human and zebrafish aCNEs, we have identified a number of conserved accessible chromatin elements that share conserved spatiotemporal, GATA-dependent activities during the early stages of heart development.

**aCNEs are enriched for lineage-specific enhancers**. Functional enrichment analysis by GREAT revealed that aCNEs were significantly associated with genes encoding DNA binding proteins, with the highest enrichment being the homeodomain proteins (GO:0043565, sequence-specific DNA binding: FDR = 0; IPR009057, Homeodomain-like: FDR = 0; binomial test). Supporting the role of aCNEs as developmental enhancers, we

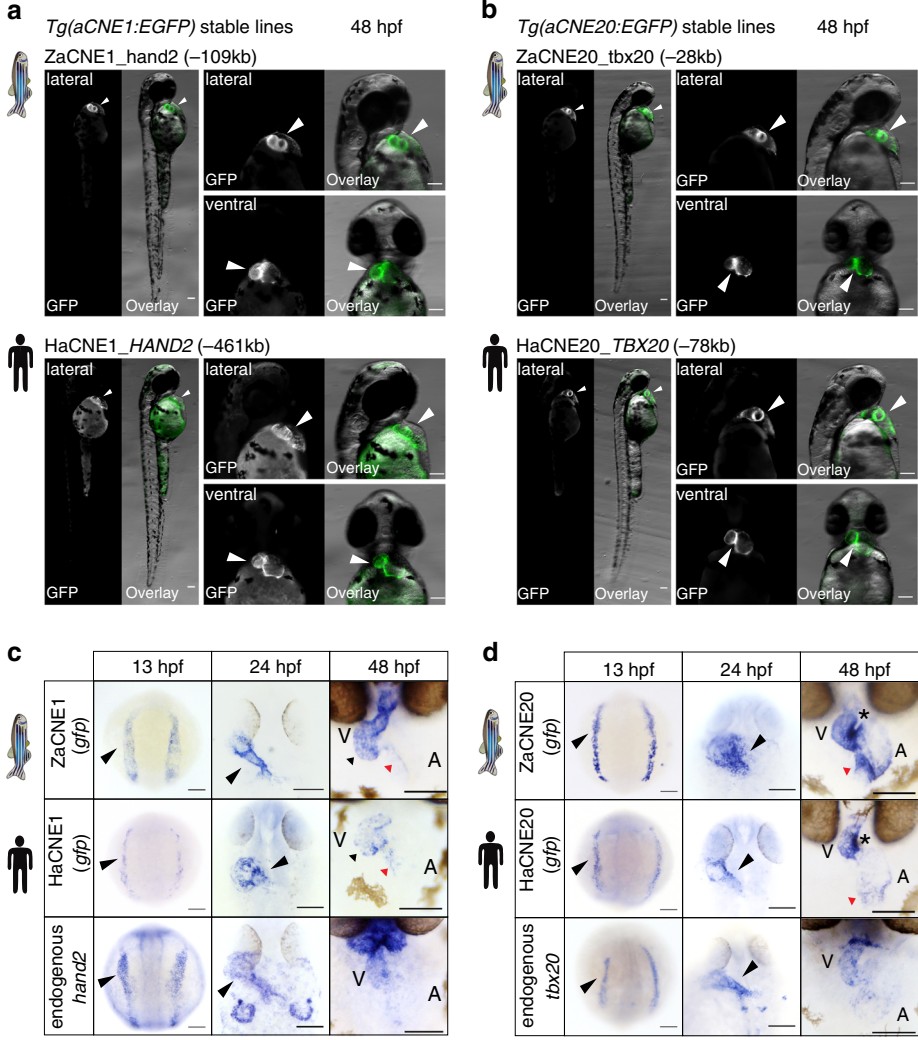

**Fig. 4** Anciently conserved open chromatin regions share conserved cardiac activities. Fluorescent images (**a**, **b**) of aCNE transgenic lines generated using zebrafish or human sequences. In-situ characterization (**c**, **d**) of the activities of the zebrafish (upper panel) and human (middle panel) aCNE sequences and the endogenous expression of zebrafish cardiac genes (lower panel) nearby. In 48 hpf images in (**c**), black triangles indicate staining in ventricles and red triangles staining in the inner curvature of atria for both aCNE1 transgenic lines. In 48 hpf images in (**d**), stars indicate the conserved activity of both aCNE20 enhancers at the inner curvature of ventricles and atrioventricular canal regions and red triangles point to the staining in inflow tract. All images shown were collected from embryos of stable lines and all scale bars represent 100 μm

observed that these regions were also highly enriched for genes regulated by PRC2 (MSigDB Perturbation, Set 'Suz12 targets': FDR = 0, Set 'Eed targets': FDR = 0; binomial test). To gain more insight into lineage-restricted functions of aCNEs, we compared the human aCNEs orthologous to our GFP+ specific and GFP− specific zebrafish ATAC-seq peaks. Although human aCNEs specific to the GFP+ and GFP− populations both showed significant enrichments for PRC2 target genes and homeobox transcription factors, only a minority of PRC2 regulated genes were associated with both GFP+ and GFP− aCNEs, indicating that these two sets of aCNEs may be involved in distinct developmental processes (Fig. 5a and Supplementary Fig. 7).

To characterize TF occupancy preferences within aCNEs, we used the human ENCODE TF binding sites (TFBS) dataset which contains uniformly analyzed ChIP-seq binding profiles for 161 TFs in 91 cell types (Transcription Factor ChIP-seq Clusters V3 from ENCODE). We asked if any TF showed occupancy enrichment within aCNEs compared to randomly selected open chromatin regions (Supplementary Fig. 7a). Overall, DNA binding factor enrichment in promoter and non-promoter aCNEs

were correlated ($R = 0.8$, $P < 2.2e{-}16$), with the top enriched factors largely overlapping (Supplementary Fig. 7b–d). The factors with the highest enrichment Z-scores were subunits of PRC2 (EZH2, SUZ12) (Fig. 5b and Supplementary Fig. 7a, b). Other transcriptional repressors (CTBP2, SIN3A, REST, KAP1) or dual regulators (TCF7L2 and YY1, associated with both active and repressive regulation in different contexts) were also seen among the top 20 most enriched factors (Fig. 5b and Supplementary Fig. 7a-d). We further observed significant enrichment for a wide-variety of TFs (GATA2, FOXP2, NANOG) and regulators of chromatin architecture (RAD21, CTCF) (Fig. 5b and Supplementary Fig. 7a, b) suggesting that aCNEs play diverse roles in gene regulation.

To gain a more comprehensive view of the tissue-specific usage of aCNEs, we leveraged available chromatin states of 127 human tissues/cell types that were predicted based on 5 chromatin marks (H3K4me3, H3K4me1, H3K36me3, H3K27me3, and H3K9me3) in the Roadmap Epigenomics Project[48] (Fig. 5c and Supplementary Fig. 8). We simplified the 15 chromatin states into Active, Bivalent, Polycomb repressed, Quiescent, and Other categories

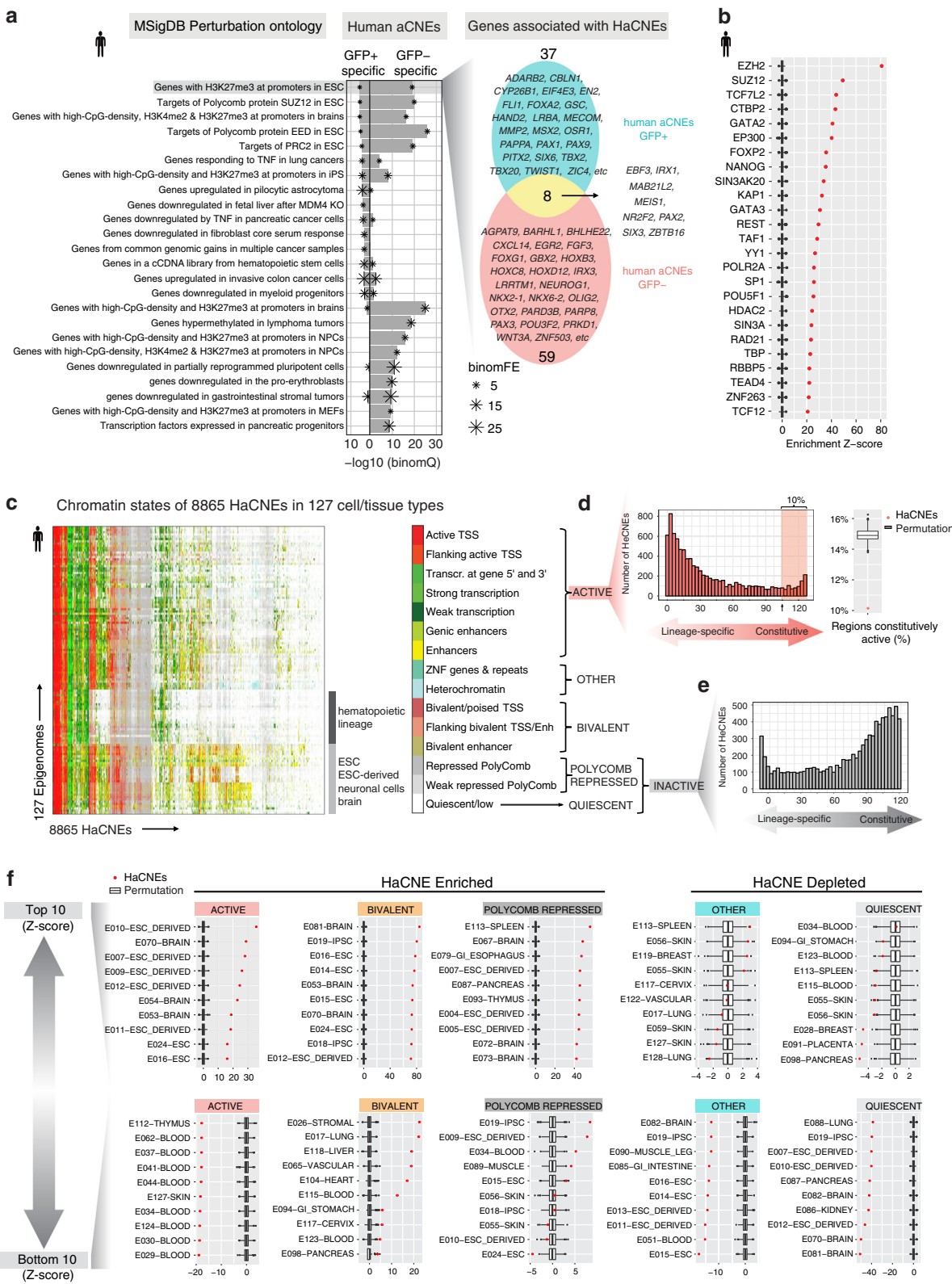

(Fig. 5c and Supplementary Fig. 8). Only 10% (899/8865) of the aCNEs showed constitutive activity (Active in >80% the tissues/cell types) (Fig. 5d). Supporting this observation, aCNEs were depleted for regions of constitutive activity compared to bulk open chromatin regions (ENCODE3 human DNase I hypersensitivity site Master list, $P = 0.001$, permutation test; Fig. 5d).

Supporting their function as lineage-restricted enhancers, we found more than half of aCNEs to be active in a lineage-specific manner (active chromatin state in at least one epigenome and Polycomb repressed or Quiescent in >70% of the epigenomes (4603/8865), Fig. 5d, e). To examine if certain chromatin states were enriched in aCNEs, we compared the percentage of a

**Fig. 5** Genomics and epigenomic features of aCNEs. **a** Enrichment analysis of HaCNEs conserved with GFP+ or GFP− specific ATAC-seq peaks. Top 15 enrichment terms (binomial FDR < 0.5, FE > 2, sorted by binomial FDR) for each category (GFP+, GFP−) were plotted. Venn diagram shows the overlap of genes associated with the GFP+ and GFP− specific HaCNEs that contribute to the top enrichment category (genes with H3K27me3 at promoters in ESC). **b** DNA binding factor occupancy enrichment within all HaCNEs ($n = 8866$). Boxplots show the distribution of 1000 times permutation and red dots represent the enrichment Z-scores normalized by the permutation results. The top 20 most enriched factors were plotted. **c** Heatmap showing the chromatin states of 8865 HaCNEs across 127 epigenomes from Roadmap Epigenomics Project. One HaCNE on chromosome Y was excluded from this analysis since it was absent in several epigenomes from female tissues. **d** Histogram showing the distribution of HaCNEs that displayed ACTIVE chromatin states (x-axis). Approximately 10% of HaCNEs were constitutively active in the majority of the epigenomes (> 80%) (shaded area in histogram). Compared to the same number of randomly selected open chromatin regions from the ENCODE3 human DHS master list, HaCNEs were depleted for regions constitutively active (red dot on boxplot on the right). **e** Histogram of HaCNEs that displayed Quiescent or Polycomb repressed states. **f** Enrichment of certain chromatin states (active, bivalent, polycomb repressed, quiescent and other) in HaCNEs compared to randomly selected open regions (permutation). The enrichment Z-scores were plotted for each epigenome (red dot) along with the permutation distribution (boxplot) (see Supplementary Methods for details). For each chromatin state, epigenomes with the top 10 (upper panel, most enriched/least depleted) and bottom 10 (lower panel, most depleted/least enriched) Z-scores were plotted. Row names were formatted as ID-EpigenomeName, which were retrieved from Roadmap Epigenomics Project metadata table (http://egg2.wustl.edu/roadmap/web_portal/meta.html). In all boxplots, center represents median, lower and higher hinges correspond to the first and third quartiles, and whiskers extend to values no further than 1.5 *(distance between first and third quantile) from the hinges. Data beyond the end of whiskers are plotted individually as outliers

chromatin state among aCNEs to that among randomly selected open chromatin regions in each epigenome. We found that aCNEs were enriched for Bivalent (126/127 epigenomes) and Polycomb repressed (121/127 epigenomes) chromatin and depleted for Quiescent (120/127 epigenomes) and Other (119/127 epigenomes) chromatin regions in the vast majority of the epigenomes (Fig. 5f and Supplementary Fig. 8c). For Active chromatin regions, aCNEs showed enrichment in some epigenomes (brain, neuronal cell types) but depletion in others (in particular hematopoietic lineages) (Fig. 5f and Supplementary Fig. 8c).

Given that lineage-specific activity and enrichment for PRC2 binding are well-established properties of poised enhancers in ESCs[49–51], we compared aCNEs to three enhancer types defined in a recent study using mouse ESCs: poised (P300+, H3K27me3+, H3K4me3−, H3K27ac−), primed (H3K4me1+, H3K4me3−, H3K27ac−, H3K27me3−) and active (P300+, H3K27ac+, H3K4me3−, H3K27me3−)[51]. Compared to all open chromatin regions, aCNEs showed a five-fold enrichment for poised enhancers (Bonferroni adjusted $P = 8.87e–19$, hypergeometric test, one-sided) with little enrichment for active (1.4 fold, Bonferroni adjusted $P = 7.21e–3$, hypergeometric test, one-sided) or primed (1.1 fold, Bonferroni adjusted $P = 1$, hypergeometric test, one-sided) enhancers. We also observed a strong enrichment for poised enhancers that were identified in human ES cells (fold of enrichment: 9.72, Bonferroni adjusted $P = 1.14e–84$, hypergeometric test, one-sided)[49].

Conserved genomic regulatory blocks (GRBs), which are defined by cluster of CNEs, contain developmental genes, coincide the boundaries of topologically associating domains (TADs) and remain concordant between species of various evolutionary distances[52]. We found that nearly 90% (7846/8866) of human aCNEs fell into GRBs defined by using CNEs (70% identity over 50 bp) between human and chicken[52], suggesting that aCNEs have been kept in a discrete set of syntenic regions during vertebrate evolution. Overall, the genomic and epigenomic properties of aCNEs suggest that many of them may serve as lineage-restricted enhancers that facilitate the expression of developmental genes.

## Discussion

The existence of a phylotypic period implies that the gene regulatory events that control phylotypic gene expression patterns are also likely to be evolutionarily constrained. In this study, we characterized a population of cells enriched for cardiac progenitors in early zebrafish embryos. By studying the open chromatin

regions in this population, we uncovered a set of putative enhancers. Leveraging two resources that characterize non-coding elements conserved between zebrafish and humans[24,25] allowed us to identify more than 6000 zebrafish open chromatin regions that overlapped open chromatin regions in human or mouse genomes. Of these conserved accessible chromatin regions (aCNEs), 162 were unique to our cardiac progenitor-cell enriched population. While 27% (43/162) of the orthologous human regions have recently been predicted to be cardiac enhancers[22], to our knowledge none of these 162 zebrafish open chromatin regions or their orthologous human or mouse sequences had been previously tested in vivo. We found that 16/18 pre-phylotypic aCNEs drove cardiac expression in stable zebrafish transgenic lines at 24 hpf, which is around the time considered to be the zebrafish phylotypic stage[3,4,7], or later in development (48 hpf). Further experiments, such as ChIP-seq for individual transcription factors and posttranslational histone modifications will help understand the spatial and temporal dynamics activity of these aCNEs. Overall, our data support the existence of conserved cis-regulatory elements that are primed early in development prior to the establishment of the body plan and function during the phylotypic stage.

Many functional CREs do not share overt sequence similarities due to rapid turnover of the spacing between, and the sequences of, TF binding motifs[39,53–55]. This phenomenon is prevalent even within rodent[56] or primate[41] orders, let alone comparisons between species separated over large evolutionary distances, such as zebrafish and human (over 400 million years from the last common ancestor). Nonetheless, sequence comparisons between human and teleosts over great phylogenetic distance have discovered many functional non-coding elements active during development[25,57–60] and CNEs are often found near genes encoding developmental regulators[24,25,57,59,61]. More direct evidence of conserved cis-regulatory events acting prior to the phylotypic period have come from profiling epigenetic changes around the phylotypic period. For example, a study using zebrafish embryos has shown that regions dynamically gaining epigenetic modification indicative of development enhancers (H3K27ac) at the gastrulation stage (8.5 hpf) are enriched for evolutionarily conserved DNA sequences[6]. Furthermore, differential DNA methylation changes observed during the phylotypic period in zebrafish, mouse, and frog were enriched for evolutionarily constrained DNA sequences and these DNA methylation changes were presumably guided by prior sequence-specific transcription binding events[7,62]. Our results identify a substantial number of aCNEs that are established in the early embryo, many

of which drive tissue-specific gene expression patterns later in development.

Vertebrate heart enhancers tend to show a lack of overt phylogenetic conservation relative to enhancers active in other tissues, such as the brain, at the same developmental stages[16,63]. Our results echo this finding, while at the same time highlighting a set of highly conserved heart enhancers. By comparing open chromatin within CNEs identified using approaches such as transitivity through a third species and ancestry reconstruction[24,25], we discovered 60% more conserved open chromatin regions between zebrafish and human/mouse that would otherwise have been missed by using direct sequence alignment alone (Fig. 3a, b). Nearly half of the heart enhancers we validated were from indirect sequence alignment (Supplementary Data 4), and zebrafish-human aCNE orthologs identified from either direct (aCNE1) or indirect (aCNE5, aCNE19, aCNE20) alignment share conserved cardiac activity to a similar degree, despite a 50–60% sequence identity. The aCNEs that we have validated can be used to further our understanding of heart development and regeneration. For example, using one of these indirectly aligned heart enhancers (aCNE21) we can recapitulate the endogenous expression of the nearby cardiac gene *hey2*[64]. Characterizing the epigenomic landscape of additional cell-lineages early in development will likely highlight further enhancers and reveal specific aCNEs capable of driving cell or tissue-specific expression before, during and after the phylotypic period.

Attesting to their potential importance in regulating specific genes through long-range chromatin interactions, an established genomic property of CNEs is that they cluster into regions of conserved synteny, referred to as gene regulatory blocks (GRBs)[65]. A significant fraction of GRB boundaries coincide with the boundaries of topological association domains (TADs)[52]. We found that nearly 90% (7846/8866) of our human aCNEs fell into GRBs comprised of CNEs between human and chicken (70% identity over 50 bp)[52]. Overall the genomic and epigenomic properties of aCNEs suggest that many of them may serve as lineage-restricted enhancers that facilitate the expression of developmental genes.

We conclude that conserved open chromatin regions established prior to the phylotypic period, and shared over 450 million years of evolution, likely represent a set of ancient enhancers that contribute to diverse spatial and temporal gene expression patterns. Although the first deletions of ultraconserved non-coding elements did not reveal overt phenotypes[66], new studies are beginning to demonstrate developmental anomalies[67,68]. Consistent with the existence of shadow enhancers, which can buffer the effects of individual enhancer loss[69–71], two recent studies showed that the pairwise deletion of ultraconserved enhancers had an increased phenotypic impact[68,71]. While more work remains to be done to dissect the in vivo, spatial-temporal expression patterns and function of anciently conserved vertebrate enhancers, regions of deeply conserved open chromatin represent a solid foundation from which the regulation and evolution of the vertebrate body plan can be explored.

## Methods

**Zebrafish husbandry and line maintenance**. Zebrafish were maintained and handled under the guidance and approval of the Canadian Council on Animal Care and the Hospital for Sick Children Laboratory Animal Services. Embryos were raised at 28.5 °C and developmentally staged by their morphology[72]. All zebrafish lines used in the study were shown in Supplementary Method.

**RNA probe synthesis and in-situ hybridization**. EGFP sequence from the ZED vector and Cre sequence from *p3E-CreERT2* were sub-cloned into pGEM Teasy vector (Promega, Cat# A1360) to make antisense probes (EGFP-F: GGATCCAT GGTGAGCAAGGGCGAGGAG, EGFP-R: CTCGAGTTACTTGTACAGCT CGTCCATGCCG; Cre-F: GGCGTTTTCTGAGCATACCTG, Cre-R: CCCAG

GCTAAGTGCCTTCTCT). DIG-labeled in-situ probes were synthesized using DIG RNA Labeling kits (Roche). RNA in-situ hybridization was carried out using the following protocol[73]. Briefly, embryos of the right stages were fixed in 4% paraformaldehyde and then dehydrated in 100% MeOH. After rehydration, embryos were permeabilized and incubated with digoxigenin-labeled antisense RNA probe. RNA-probe hybrids were detected by an alkaline phosphatase-conjugated antibody (Anti-Digoxigenin-AP, Fab Fragments, 1:5000, Roche, Cat# 11093274910) that catalyzed reaction on a chromogenic substrate. Stained embryos were cleared in BBA solution (2:1 Benzyl benzoate: Benzyl alcohol) and imaged under a Leica M205FA stereomicroscope

**Immunostaining**. Embryos from *Tg (Smarcd3-F6: EGFP)*[hsc70] and *Tg(nkx2.5: ZsYellow)*[fb7] crosses were fixed at 13 hpf in 4% paraformaldehyde at 4 °C overnight. After 3 × 5 min washes in PBS with 0.1% Triton, embryos were permeabilized in PBS with 0.5% Triton for 4 h at room temperature (RT). Embryos were then blocked in PBST (1% DMSO and 0.5% Triton X in PBS) with 5% Normal Goat Serum (Millipore, Cat# S26-LITER) for 1.5–2 h at RT before being incubated with primary antibodies (α-RCFP 1:500, Clontech Cat# 632475; α-GFP 1:1000, ThermoFisher Cat# A-11120) at 4 °C overnight. The Next day, embryos were washed for 3–4 h in PBS with 0.1% Triton at RT, with 6–8 changes of solution. Incubation of secondary antibodies (α-mouse IgG-FITC, 1:1000, Santa Cruz Cat# SC-2010; α-rabbit IgG, Alex 568, 1:1000, ThermoFisher Cat# A-1101) was carried out at 4 °C overnight, followed by the same washing procedure as that for primary antibodies. After staining, embryos were mounted in 1% (w/v) low melt agarose (Sigma, A9414) and imaged under a Nikon A1R Si Point Scanning Confocal microscope.

**CreETR2 lineage tracing**. Embryos from *Tg (Smarcd3-F6:CreERT2)*[hsc76] and *Tg (βactin2:loxP-DsRed-STOP-loxP-EGFP)*[s928Tg] crosses were dechorionated and incubated in 5 μM 4-OH-Tamoxifen(4-HT, Sigma cat# T176) for 12 h. The time of 4-HT addition was specified in Supplementary Fig. 1. After treatment, embryos were rinsed twice, placed in fresh egg water and imaged at 48 hpf under a Zeiss Axio Zoom.V16 Stereoscope.

**Embryo dissociation, fluorescence-activated cell sorting**. Around 100 *Tg (Smarcd3-F6: EGFP)*[hsc70] embryos were dechorionated with pronase (Sigma, Cat# 11459643001) and transferred to a 1.5 mL Eppendorf tube when they reached the bud stage. After incubation in 200 μl calcium-free Ringer solution (116 mM NaCl, 2.6 mM KCl, 5 mM HEPE, pH 7.0) for 5 min, embryos were transferred into a 24-well plate with 500 μl TrypLE solution (GIBOCO, TrypLE Express Enzyme, cat #: 12604-013) for dissociation at room temperature. Embryos were gently homogenized every 5 min with P1000 tips. Dissociation was monitored under a dissection scope until most cells were in a single-cell suspension. The cell suspension was transferred into 200 μl ice-cold fetal bovine serum (FBS) to stop the reaction. Cells were centrifuged at 300 g for 3 min at 4 °C and washed with 500 μl ice-cold DMEM with 10% FBS before resuspended in 500 μl ice-cold DMEM with 1% FBS. Right before the Fluorescence-activated cell sorting (FACS), cells were filtered through a 40 μm strainer and DAPI was added at a concentration of 5 μg/ml to exclude dead cells.

FACS was performed on Beckman Coulter Mo Flo XDP or Mo Flo Astrios sorter, or Sony SH800S Cell Sorter with a 100 μm nozzle by the SickKids-UHN Flow and Mass Cytometry Facility. Cell-doublets and dead cells were excluded based on forward scatter, side scatter and DAPI channel. GFP+ and GFP− cells were sorted into 100% FBS and subjected to RNA-seq or ATAC-seq procedures immediately after sorting. Approximately 30,000–50,000 GFP+ cells and 100,000 GFP− cells were collected in one FACS run.

**Bulk mRNA-seq and single-cell mRNA-seq**. Single-cell cDNA libraries were prepared using Fluidigm C1 system. After FACS, GFP+ cells were washed twice in DMEM with 3% FBS and filtered through a 40 μm cell strainer. Cells were adjusted to a concentration of 400–500 cells/μl and mixed with C1 suspension solution at a 5.2:4.8 ratio. 10 μl of the final cell mixture was loaded into a C1 medium or small Chip. Cell capture was examined under a microscope and only wells with a single cell captured were included in library construction. 3 ArrayControl™ RNA Spikes were added into the cell lysis mixture according to the Fluidigm C1 single-cell mRNA-seq protocol (PN 100–7168 Rev. B1). Cell lysis, Oligo-dT primer mediated reverse transcription, 21 cycles of PCR amplification and cDNA harvesting were performed as per manufacturer's instruction (Fluidigm, PN 100–7168 Rev. B1). We usually recovered 30–40 cells (30–40% capture efficiency) from one C1 Chip and 96 single-cell cDNA libraries were collected from three batches of experiments.

For bulk mRNA-seq, RNA from 4000 GFP+ or GFP− cells were prepared using RNeasy Plus Micro Kit (Qiagen, Cat# 74034) and cDNA libraries were prepared following the Fluidigm tube control protocol (the same protocol as that for single-cell mRNA-seq except the input cell number is different). Three biological replicates were collected for both GFP+ and GFP− samples.

For both single-cell and bulk mRNA-seq, final sequencing libraries were made using Nextera XT DNA Sample Preparation Kit (Illumina, Cat# FC-131–1096) and 120 bp pair-end sequenced on an Illumina HiSeq 2500 platform according to manufacturer's instruction. Bulk RNA-seq libraries were sequenced to a depth of

(18 ± 1.9) million reads and single-cell mRNA-seq libraries a depth of (3.0 ± 0.7) million reads.

**ATAC-seq**. ATAC-seq was performed with minor modifications to the published protocol[74]. 30,000–50,000 cells obtained from FACS were used for nuclei prep. After tagmentation, transposed DNA fragments were amplified using the following PCR condition (1 cycle of 72 °C for 5 min and 98 °C for 30 s, followed by 12 cycles of 98 °C for 10 s, 63 °C for 30 s and 72 °C for 1 min). Amplified libraries were purified twice with Agencourt Ampure XP beads (Beckman Coulter, Cat#A63880) with a bead-to-sample ratio of 1.8:1. A Qubit fluorometer and Aglient Bioanalyzer were used to check library quality and concentration. Libraries were 50 bp single-end sequenced on Illumina HiSeq 2500 platform to a depth of $(3.5–7.0) \times 10^7$ reads. Two biological replicates were collected for both GFP+ and GFP− cells.

**Transgenic zebrafish enhancer assay**. Candidate regions containing the zebrafish ATAC-seq peaks (21 ZaCNEs) or human DHSs (4 HaCNEs) were amplified from genomic DNA and recombined into *pDONOR221* vector (Invitrogen, Gateway BP Clonase II Enzyme Mix, Cat# 11789020) before they were cloned into *E1b-Tol2-GFP-gw* vector. 25 ng *E1b-Tol2-GFP-gw* plasmid carrying one aCNE and 150 ng Tol2 mRNA were injected into wild-type embryos at one-cell stage. F0 founder embryos were raised to 48–52 hpf before imaging and heart expression scoring. Candidate regions were considered as a heart enhancer if more than 30% of the injected embryos displayed GFP+ cells beating in the hearts, which was consistent with the criterion used in similar studies before[75]. 42–220 (average n = 86) injected embryos were analyzed for each candidate region. The genomic coordinates, lengths and nearby genes of candidate regions and primers used for cloning can be seen in the Supplementary Data 4.

F0 embryos injected with enhancers that passed the 30% threshold were raised to screen for transgene germline carriers. Except enhancer ZaCNE18 for which only one carrier has been identified, 2–4 independent alleles have been identified for each enhancer (see Supplementary Data 4). Though ectopic expression has been seen in some carriers, the cardiac expression patterns were similar in different alleles of the same transgene.

**GATA motif mutagenesis**. GATA motif mutation was introduced by primers designed by Agilent Genomics QuickChange program (http://www.genomics.agilent.com/primerDesignProgram.jsp) (ZaCNE1_muta_F: cagattaggacccagctaggtgccagtgggggggggtgttagtgcagaaaaggttacactac; ZaCNE1_muta_R: gtagtgtaacctttctgcactaacacccccccccactggcacctagctgggtcctaatctg; HaCNE1_muta_F: attagagtgaaaagaggtgccggtgggggggggtgcgaatgcgccagggtcacgc; HaCNE1_muta_R: gcgtgaccctggcgcattcgcaccccccccaccggcacctcttttcactctaat).

The primers were designed to convert the aligned GATA consensus sequence AGATAA to CCCCCC. *pDONOR221* vectors carrying the mutated aCNE1 enhancers were PCR amplified from the original *pDONOR221* containing the wild-type aCNE1 sequences using the primers above. After amplification, 50 μl PCR mixture was incubated with 1 μl DpnI (NEB, Cat#R0176S) at 37 °C to remove wild-type enhancer templates. DpnI was inactivated at 80 °C for 20 min before transformation. Plasmid clones with the correct GATA motif mutation were confirmed by Sanger sequencing and used as entry clones to make *E1b-Tol2-GFP* constructs containing mutated aCNE1 enhancers. Independent germline carriers have been identified for ZaCNE1_GATAMutated (n = 5) and HaCNE1_GATAMutated (*n* = 4) enhancers (see Supplementary Data 4). GFP expression levels in different GATA_Mutated alleles look slightly different but were generally weaker than wildtype_alleles. Images in Supplementary Fig. 6 were taken using alleles showing median expression level within all alleles.

**Analyses of mRNA-seq and ATAC-seq data**. Analysis pipelines of mRNA-seq and ATAC-seq data are shown in Supplementary Methods.

**Identification of sequence-conserved open regions**. Two zebrafish CNE datasets were used for this analysis. First, Zebrafish CNEs (zCNEs) conserved with human or mouse identified from Hiller et al.[24] were obtained from the authors, including records of direct sequence alignment and transitive mapping through other species or reconstructed ancestries[24]. zCNEs that overlaps a zebrafish-human well-aligning window[24] by at least 15 bp were defined as directly aligned. zCNEs that can be mapped back to mouse through transitive alignment or ancestry reconstruction but cannot be detected by direct alignment were defined as indirectly aligned CNEs. The same direct and indirect alignment definition was set for zCNE conserved with mouse (zCNE_mouse)

To include more high-quality zebrafish CNEs into our analysis, we used another zebrafish-human CNE dataset identified in a recent study through transitive alignment via the spotted gar genome[25], which we referred to as garCNE. If the zebrafish coordinate of a garCNE does not overlap any zCNE records, we added it into our analysis and define it as indirectly aligned. Altogether, we collect 20,005 zebrafish CNEs conserved with human, with 10187 direct and 9818 indirect ones (Supplementary Table 1).

To establish the same system for CNEs conserved between zebrafish and mouse, we used liftOver to convert the human-zebrafish CNEs identified through gar

genome to mouse-zebrafish CNEs (-minMatch = 0.1), and then add those CNEs on top of zCNE_mouse dataset in a similar way as what we did for human. At the end we obtained 18,827 zebrafish CNEs conserved with mouse, with 9429 direct and 9399 indirect ones (Supplementary Table 1). Finally, we associated zebrafish ATAC-seq peaks to CNEs (sequence-conserved ATAC-seq peaks) if they overlap a CNE by at least one base pair. Nearly 90% of the these CNE-associated ATAC-seq peaks were completely excluded from gene coding regions defined by Ensembl transcriptome annotation (Zv9, release 79).

**Identification of aCNEs**. Human or mouse DNase master peak lists from ENCODE3[42] (hg19: ENCFF257KON; mm10: ENCFF203SGZ) were used for identifying open regions anciently conserved between zebrafish and human or between zebrafish and mouse. Mouse DNase master peaks, which were provided as mm10 coordinates, were converted to mm9 coordinates using liftOver (-minMatch = 0.95). If a CNE that overlaps a zebrafish ATAC-seq peak by at least one base pair also overlaps a DNase I hypersensitivity site (DHS) from the DNase master lists by at least one base pair, then these orthologous ATAC-seq peak and DHS linked via CNEs are identified as accessible CNEs (aCNEs).

There are 1,199,722 non-overlapping regions in mouse DHSs master list after liftOver, 4,193,929 non-overlapping regions in human DHS master list (Supplementary Table 3). Probably due to the total number differences, more human DHSs have been identified as aCNEs than mouse DHSs; similarly, more ATAC-seq peaks were identified as aCNEs shared with human than with mouse (Supplementary Table 3).

We found analyses with the DHS master lists sometimes gave us a chain of DHSs that are conserved with the same ATAC-seq peaks. To avoid potential bias that may be introduced in analysis conducted by GREAT, we merged the DHSs that are within 200 bp and conserved with the same ATAC-seq peaks. We used merged DHSs for analyses in Fig. 5a, Supplementary Figs. 3f, 7b, c. All other analyses were conducted on original DHS coordinates.

Comparative genomic analyses of aCNEs are shown in Supplementary Methods.

## Data availability
ATAC-seq and RNA-seq data are available at ArrayExpress under the accession number E-MTAB-6078 and E-MTAB-6077. All other relevant data supporting the key findings of this study are available within the article and its Supplementary Information files or from the corresponding author upon reasonable request.

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

## Acknowledgements

We would like to thank all members of the Wilson and Scott Labs for helpful discussion and suggestions. Special thanks go to: Huayun Hou and Minggao Liang for guidance and comments on computational analyses; Angel Morley, Allen Ng, Scott Knox and Alejandro Salazar for expert fish husbandry; Michelle Ward and Sara Good for helpful comments on the manuscript; Michael Hiller for sharing the zCNEs full dataset and providing a detailed explanation regarding the dataset. Transgenic lines were kindly provided by Caroline and Geoffrey Burns (Tg(nkx2.5: ZsYellow) *fb7*); and Ryan Anderson (Tg(βactin2:loxP-DsRed-STOP-loxP-EGFP)*s928Tg*). Expert help was provided by the SickKids-UHN Flow and Mass Cytometry Facility; and Sergio Pereira from The Center

for Applied Genomics (TCAG) facility (bulk and single-cell mRNA-seq). We thank the ENCODE and Roadmap Epigenomics Mapping Consortium for generating and consolidating the open chromatin landscapes and chromatin state datasets. This research was supported by Hospital for Sick Children Restracomp Studentship (to X.Y.) and Connaught International Scholarship (to X.Y.); Ontario Trillium Scholarship (to M.S.); Labatt Family Heart Centre for Innovation Funds to I.C.S. and M.D.W; Heart and Stroke Foundation of Canada (to I.C.S. and M.D.W., Grant-in-Aid G-16-00013798); NHBLI R01HL114948 (to B.B.); and the Natural Sciences and Engineering Research Council of Canada (NSERC) grant 436194-2013 (to M.D.W.). M.D.W. is supported by a Canada Research Chair (CRC) in Comparative Genomics and Early Researcher Awards from the Ontario Ministry of Research, Innovation and Science.

## Author contributions

Conception and study design: X.Y., B.B., I.C.S., and M.D.W.; Acquisition of data: X.Y., M.S., and P.D.; Analysis and interpretation of data: X.Y., I.C.S., and M.D.W.; supervision of work: M.D.W., I.C.S., and B.B.; X.Y., I.C.S., and M.D.W. wrote the manuscript and all authors assisted with drafting and revision of the manuscript.

## Additional information

**Competing interests:** The authors declare no competing interests.

