## [Peer Review File · Nature Communications]

Reviewer #1 (Remarks to the Author):

This is an elegant and compelling study showing that ancient conserved noncoding elements (aCNEs) identified as nuclease-accessible regions in cardiac progenitor cells are indeed active as enhancers during early stages of cardiac determination. The authors use an enhancer active very early in cardiogenesis (Smarcd3-F6, active before the classic heart transcription factor NFX25 is produced) to drive a GFP reporter to mark early cardiac progenitors in zebrafish. ATAC-seq was then used to find nuclease-accessible regions, a hallmark of cis-regulatory elements (CREs). From the set of about 200,000 ATAC-seq peaks, about 3800 were specific to GFP+ cells and about 1600 were in GFP- cells. The authors used sets of conserved noncoding elements (CNEs) found in both humans and fish (or mouse and fish) to identify the ATAC-seq peaks specific to early cardiac progenitors that are also conserved as nuclease accessible regions across these species. Notably, they employed both direct assessments (mammalian-fish alignments) or indirect assessments (leveraging alignment to an intermediate species) to improve the sensitivity of the analysis. The authors then show that these deeply preserved nuclease-accessible regions (aCNEs) act as enhancers driving expression in the developing heart. They mined the large amount of information on mammalian noncoding sequences active as enhancers during mouse development. In addition, they tested a reasonable number of the aCNEs in transgenic zebrafish, showing that a substantial fraction are active in the embryonic heart. The human and zebrafish orthologous aCNEs drove expression in similar spatial and temporal patterns. The authors also describe a larger set of aCNEs genome-wide, not limited to cardiac progenitors, and show that these are enriched for poised enhancers. By analogy to those studied in cardiac development, many of these may serve as early developmental enhancers.

This report is a strong advance in our understanding of the subset of CREs that are deeply preserved in evolution, showing that many of them are regulating key genes in early embryonic development. Furthermore, this study illuminates more thoroughly the conserved regulatory elements that can account for the similarities in gene expression patterns during the phylotypic period of differentiation.

The design of the experiments is sound, the data strongly support the conclusions, and the report is clear. The text is somewhat long, but that is because it covers many analyses, not because it is redundant.

These minor issues need to be corrected.

(1) Numbers of ATAC-seq peaks are not consistent from p. 7 to p. 8. Are the total number of peaks about 150,000 or are there almost 200,000 common peaks?

(2) p. 9, line 15: The word "specific" needs to be added to say "176 were GFP+ specific, 264 were GFP- specific".

(3) p. 12, line 6: If 18 of the 21 tested regions drove heart expression, then the size of the set of enhancers is 18, not 21.

(4) p. 17, last paragraph before Discussion: "discrete" not "discreet"

Consistent with the transparent review process of Nature Communications, I am happy to let people know my identity,

Ross Hardison

Reviewer #2 (Remarks to the Author):

The study by Yuan and coworkers focuses on the identification of enhancers that are conserved in the heart of zebrafish, mouse and men. They use a non-mammalian zebrafish transgene to label cardiac progenitor cells. ATAC and RNA-seq was used to identify accessible chromatin regions and gene expression signatures. The authors compare these data with previously published data from mouse and human hearts. Out of 3838 cardiac progenitor specific-sites the authors found 162 to be conserved between species. The regulatory potential of 16 representative enhancer was confirmed using a reporter construct in zebrafish. Overall this manuscript nicely shows the degree of conservation of enhancers in the heart between fish and mammals.

Concerns:

1) The authors found 3838 ATAC peaks to be specific for cardiac progenitor cells. Of those a relevant fraction (approx. 10%) overlap promoter regions. Since the manuscript focuses on enhancer regions, the authors should exclude peaks overlapping promoter regions from all subsequent analysis steps.

The authors identify only a small fraction (10-15%) of ATAC peaks in GFP+ showing the histone signature of active enhancers in zebrafish heart (S3A, K27ac+,4me1+,4me3-). Maybe the discrepancy between histone signature and ATAC peaks is the explanation for the low fraction of conserved regions.

Do all annotated ATAC peaks really represent active enhancers?

Or does the peak calling identify nucleosome positioning-associated open chromatin not directly linked to transcription factor activity?

A valid identification of enhancers is of central importance for this manuscript. Therefore, the authors should identify enhancers in GFP+ cells using a second independent method (p300, K27ac, DNA methylation).

Please add the ATAC peak annotation to all representative traces.

2) Also for the comparative genome analysis using the entire set of ATAC peaks, the authors should clearly discriminate promoter and distal regions. It is expected, that the conservation of the sequence and chromatin state of promoter and coding regions is higher as compared to enhancers. It would be of great interest to display the conservation of promoters and enhancers side by side.

3) At many places the manuscript lacks statistical information, i.e. is the overlap with the VISTA data significant? The authors should consequently include random genomic regions with a comparable size distribution in the analysis to be able to test the significance and degree of conservation.

4) The heart is a heterocellular organ and the main cell types arise from a common cardiac progenitor. A large fraction of genes, TFs and gene ontology terms (GREAT) linked to conserved regions in the heart are cardiomyocyte specific. Is the conservation of enhancer elements in the heart different between cell types? The authors should reanalyze existing human and murine cardiomyocyte data in comparison to heart tissue to address this question.

5) Some sentences are not logic. For example: Of these 6294 ATAC-seq peaks 176 were GFP+.

6) Please use a different type of Venn diagram (Fig. S3a), which allows to identify the individual intersections more easily.

Reviewer #3 (Remarks to the Author):

In the manuscript entitled “Heart enhancers with deeply conserved regulatory activity are established early in development” Yuan and colleagues perform comparative epigenomic analysis to identify evolutionarily conserved cardiac enhancer. They introduced the mouse *smarcd3-F6* enhancer, the enhancer of early specified cardiac precursors, into zebrafish and established transgenic animal for early cardiac precursor specific RNA-seq and ATAC-seq experiments. Exploiting an impressive amount of transgenic tools, they identify multiple cardiac enhancers, some of which are well-conserved in both zebrafish and human and show activity in zebrafish heart. The authors

also analyzed usages of a large number of conserved accessible chromatin (aCNEs) in 127 human tissue/cell types.

Overall, this is an exciting study to understand the function of the conserved sequences between fish and mammal. Introducing mammalian enhancer into zebrafish and determining conserved enhancers across species is a conceptually well-designed approach. Their bioinformatics analysis can apply to other projects and can provide a new direction to identify novel enhancers. The total number of established transgenic lines are truly impressive, and these lines can be valuable resources to elucidate gene regulatory networks underlying early cardiac development.

Despite these strengths, there are several areas in which this manuscript could be significantly improved.

Major Comments

1. Enhancer activity of the smarcd3-F6:EGFP transgenic line

What is native EGFP expression without using immunostaining or in situ hybridization (ISH)? The authors FACS-sorted with EGFP signal, suggesting that EGFP can be observed without further staining. The authors should provide a figure of native EGFP expression to demonstrate which cells are sorted for ATAC-seq and RNA-seq. In addition, the authors need to briefly mention why they detected enhancer activity by ISH and immunostaining rather than live EGFP imaging.

2. Figure 2. D. The authors performed a known-down experiment of gata5/6 to examine whether smarcd3-F6 enhancer activity requires the function of Gata5 and Gata6. However, they should perform the following control experiments. 1) Known-down of sox gene(s) and then examine EGFP expression: This experiment will confirm that disruption of sox gene, binding motifs of which are enriched in ATAC-seq profiles of GFP negative cell, would not affect the activity of smarcd3-F6 enhancer. 2) Expression of ectoderm gene(s) in gata5/6 KD embryos. emilin3a, nkx2.9, or lhx2b would be used as an ectoderm gene.

3. ZaCNE and HaCNE transgenic lines

Transgenic zebrafish lines generated with ZaCNEs and HaCNEs are valuable reagents to elucidate novel sub-population of cardiac progenitor cells and related cardiac TF motif characterization in cardiac enhancers. For improving their analysis, the author should perform the following experiments:

1) Expression patterns in early time point

In Discussion, Page 18, line 9, the authors described that they tested enhancer activity in stable zebrafish lines at 24 hpf. However, this data is missing. All EGFP expressions in figure 3D, 4A, 4B, Sup. Fig. 5, and Sup. Fig. 6 are taken at 48 hpf, not 24 hpf. The authors should provide 24 hpf images, which are more valuable than 48 hpf. 48 hpf does not represent the early developmental stage (the zebrafish phylotypic stage), which this paper addresses. All enhancers are identified using 10 hpf cardiac progenitor cells, so I assume that EGFP expression can be detected by 24 hpf without additional staining.

2) I am wondering why the authors again use ISH to confirm enhancer activity at the earlier time point such as 24 hpf. They need to briefly mention why they used ISH to avoid the reader's confusion that ISH is likely a common method to detect enhancer activity with EGFP tg lines.

3) Activity of HaCNEs: The authors generate stable transgenic lines with 4 HaCNEs, but their expression patterns are not categorized. The authors need to provide what kind of category enhancers they are. If HaCNEs exhibit different EGFP expression patterns compared to their corresponding zebrafish enhancers, it would be interesting since analyzing sequence differences between ZaCNE and HaCNE can reveal the important cardiac motifs.

4) The authors described that they found 4 major categories with 17 lines. Is there any characteristic motif composition representing each category? It would be good to present maps highlighted cardiac TF motifs for at least 8 CNE (ZaCNE1, ZaCNE5, ZaCNE19, ZaCNE20, HaCNE1, HaCNE5, HaCNE19, and HaCNE20). Where are GATA4, TBX5, NKX2.5, and/or HAND2 motifs in each enhancer? How far are these motifs apart from each other in both zebrafish and human enhancers? How many copies of each motif? This analysis may elucidate novel signatures of cardiac progenitor sub-population.

4. Figure legend 5

Current Figure Legend 5 is not related. The authors must change it. For instance, (A) does not indicate Heatmap. It is about GREAT enrichment analysis.

Minor comments

1. Supplementary figure 1, Figure legend (A)

Superscript 1 and 2 would indicate references, but I couldn't find it. I believe that the authors will provide references used for supplementary figures in the final version. In addition, I could not find how the authors get the H3K27ac data of MES, CP, and CM. In the Method of "Processing and analyzing of published mouse cardiac TF ChIP-seq data", there is no description of these profiles. In addition, Method does not include any H3K27ac ChIP-seq analysis.

2. Figure 1 and page 6 line 3

To eliminate the possible effect of the maternally derived mRNA, in situ hybridization image of very early time point embryos are required (earlier than 6 hpf).

3. Page 6 line 8 – 9 "Co-immunostaining comparing Tg(Smarcd3-F6:EGFP) and Tg(nkx2.5:ZsYellow) expression indicated that the Smarcd3-F6 enhancer marked all cardiac mesoderm expressing nkx2.5 at early somite stages (13 hpf) (Fig. 1C)."

Although most nkx2.5:ZsYellow positive cells are labeled by EGFP, there are some ZsYellow positive and smarcd3-F6:EGFP negative cells. Thus, "all cardiac mesoderm" should be toned down (for example "almost").

4. Page 9 line 10

At least one base pair -> it could be fifteen.

5. Page 12, Second paragraph, line 7, 10

Experimentally determined binding sites:

There are no descriptions what these binding sites are. Only referring references are not enough. The authors should mention the exact sequences of "experimentally determined binding sites" in the supplementary data.

6. Figure 3D and supplementary figure 5

It would be good to make another column for category (expression pattern) in supplementary table 3.

7. Page 15, line 16: Fig 3D -> Fig. 5B

8. Page 16, line 4: the the -> remove one the

9. Sup. Fig. 4, 5: There is no annotation about time point of embryos they used for imaging. I assume 48 hpf. The authors should annotate the time point.

10. Method: No data about antibodies.

Reviewers' comments:

Reviewer #1 (Remarks to the Author):

This report is a strong advance in our understanding of the subset of CREs that are deeply preserved in evolution, showing that many of them are regulating key genes in early embryonic development. Furthermore, this study illuminates more thoroughly the conserved regulatory elements that can account for the similarities in gene expression patterns during the phylotypic period of differentiation.

The design of the experiments is sound, the data strongly support the conclusions, and the report is clear. The text is somewhat long, but that is because it covers many analyses, not because it is redundant.

We thank Dr. Hardison for this thoughtful review, very kind words as well as noticing specific issues we overlooked.

These minor issues need to be corrected.

(1) Numbers of ATAC-seq peaks are not consistent from p. 7 to p. 8. Are the total number of peaks about 150,000 or are there almost 200,000 common peaks?

Thank you for pointing out this confusing description. We have clarified the number of peaks and how they were generated in the text.

Changes made in main text regarding this comment:

Page 7-8: “Using ATAC-seq, we detected 155,879 open chromatin regions (ATAC-seq peaks) in the GFP+ population and 153,777 in the GFP- population. After conducting differential analysis, we found most ATAC-seq peak regions (n=195,466) shared similar ATAC-seq signals in both GFP+ and GFP- populations. 5,471 peaks showed significant quantitative differences (Fig. 2A) with 3,838 peaks specifically increased in *Smarca3*-F6 labeled cells (‘GFP+ specific’), and 1,633 peaks specifically increased in unlabeled cells (‘GFP- specific’).”

Page 31-32 (Methods): “DiffBind first generated a consensus peak set that represent an overall set of candidate open chromatin regions based on all four samples that we fed in (both GFP+ and GFP- samples). Then it counted if the sequence mapped to each interval in the consensus peak set were significantly different in GFP+ and GFP- samples. After DiffBind analysis, we obtained a total of 200,937 ATAC-seq peaks (consensus peak set), within which 3838 were GFP⁺ specific peaks and 1633 were GFP⁻ specific. We used the output from DiffBind for downstream analysis.”

(2) p. 9, line 15: The word "specific" needs to be added to say "176 were GFP+ specific, 264 were GFP- specific".

This has been corrected

(3) p. 12, line 6: If 18 of the 21 tested regions drove heart expression, then the size of the set of enhancers is 18, not 21.

This has been corrected

(4) p. 17, last paragraph before Discussion: "discrete" not "discreet"

This has been corrected

Reviewer #2 (Remarks to the Author):

The study by Yuan and coworkers focuses on the identification of enhancers that are conserved in the heart of zebrafish, mouse and men. They use a non-mammalian zebrafish transgene to label cardiac progenitor cells. ATAC and RNA-seq was used to identify accessible chromatin regions and gene expression signatures. The authors compare these data with previously published data from mouse and human hearts. Out of 3838 cardiac progenitor specific-sites the authors found 162 to be conserved between species. The regulatory potential of 16 representative enhancer was confirmed using a reporter construct in zebrafish. Overall this manuscript nicely shows the degree of conservation of enhancers in the heart between fish and mammals.

We thank you for this appraisal and for your thoughtful comments.

Concerns:

1) The authors found 3838 ATAC peaks to be specific for cardiac progenitor cells. Of those a relevant fraction (approx. 10%) overlap promoter regions. Since the manuscript focuses on enhancer regions, the authors should exclude peaks overlapping promoter regions from all subsequent analysis steps.

We agree with the reviewer in the value of considering the promoter and enhancer regions separately. Please see our comments for the more specific and related question in concern 2 below. We have now included several analyses comparing the genomic properties at proximal promoter and distal regions (See Reviewer 2 – concern #2 for details). Since our paper focused on accessible conserved non-coding elements we still see value in including all such regions in our analysis.

The authors identify only a small fraction (10-15%) of ATAC peaks in GFP+ showing the histone signature of active enhancers in zebrafish heart (S3A, K27ac+,4me1+,4me3-). Maybe the discrepancy between histone signature and ATAC peaks is the explanation for the low fraction of conserved regions. Do all annotated ATAC peaks really represent active enhancers? Or does the peak calling identify nucleosome positioning-associated open chromatin not directly linked to transcription factor activity?

We thank the reviewer for bringing up this point. Firstly, we do not believe or claim that all ATAC-seq peaks are active enhancers, and yes there is clear evidence that methods such as DNase-seq (1) and ATAC-seq (2) occur at areas of the genome lacking active chromatin marks (e.g. The ENCODE consortium Thurman et al. 2012, described more than 2.4 million DNase hypersensitivity sites in 112 samples (1)). While the enhancer function of the vast majority of CTCF sites has not been tested, CTCF binding sites represent well studied examples of accessible chromatin regions bound by TFs that don't have enhancer function (see (2) for one of the early studies looking at this genome wide). An important study looking explicitly at DNase I sites that don't overlap active chromatin marks was published by Ernst and Kellis in 2013(4). Results from this study are relevant to our work and so we have added text to the results to address these concerns and properly acknowledge this previous pioneering work (page 8):

“We found that 69% of accessible zebrafish regions did not overlap with active chromatin marks (Supplementary Fig. 3A). The prevalence of open chromatin regions with low levels of active chromatin marks has been readily observed in human cell lines and associated with a signature of motif-dependent binding characteristic of repressive chromatin states (4). It is likely that the lack of overlap of our accessible chromatin regions with active chromatin marks obtained from whole embryos will involve such regions in addition to being due to technical reasons, such as having purified a small subset of cells away from the bulk embryo prior to ATAC-seq.”

(1)Thurman RE, et al. The accessible chromatin landscape of the human genome. *Nature*. 2012 Sep 6;489(7414):75-82. doi: 10.1038/nature11232. PMID: 22955617

(2) Buenrostro JD, Giresi PG, Zaba LC, Chang HY, Greenleaf WJ. Transposition of native chromatin for fast and sensitive epigenomic profiling of open chromatin, DNA-binding proteins and nucleosome position. *Nat Methods*. 2013 Dec;10(12):1213-8. doi: 10.1038/nmeth.2688. Epub 2013 Oct 6. PMID: 24097267;

(3) Song L, et al. Open chromatin defined by DNaseI and FAIRE identifies regulatory elements that shape cell-type identity. *Genome Res*. 2011 Oct;21(10):1757-67. doi: 10.1101/gr.121541.111. Epub 2011 Jul 12. PMID: 21750106;

(4) Ernst J, Kellis M. Interplay between chromatin state, regulator binding, and regulatory motifs in six human cell types. *Genome Res*. 2013 Jul;23(7):1142-54. doi: 10.1101/gr.144840.112. Epub 2013 Apr 17. PMID: 23595227

A valid identification of enhancers is of central importance for this manuscript. Therefore, the authors should identify enhancers in GFP+ cells using a second independent method (p300, K27ac, DNA methylation).

We agree with the reviewer that discovering enhancers is of central importance to this work. We believe the strength of our work comes from testing the enhancer activity of a subset of these ATAC-seq peaks using additional in vivo assays. Furthermore, by being able to map zebrafish ATAC-seq peaks to ~8K human open chromatin regions, we have been able to obtain a much richer and varied view of the putative enhancer activity of aCNE from a human epigenome (Thanks to ENCODE and the Roadmap Epigenome Project (see Fig. 5, Supplemental Fig. 7 and 8). While performing additional ChIP-seq or DNA methylation assays is zebrafish of great interest to us, these experiments are beyond the scope of this work.

Please add the ATAC peak annotation to all representative traces.

We tried labelling the individual ATAC-seq peaks in Fig. 2A however we found it was clearer just to give three specific examples. So to address this comment we have now included a full list of processed and annotated ATAC-seq peaks that people can view themselves (see Supplemental Table 1).

2) Also for the comparative genome analysis using the entire set of ATAC peaks, the authors should clearly discriminate promoter and distal regions. It is expected, that the conservation of the sequence and chromatin state of promoter and coding regions is higher as compared to enhancers. It would be of great interest to display the conservation of promoters and enhancers side by side.

This is a great suggestion. We have now separated promoter and non-promoter aCNEs and reanalyzed the following genomic features:

- GC content, open chromatin signal and boundary (added to Supplementary Fig. 3)
- TF binding enrichment (added to Supplementary Fig. 7)

Although ATAC-seq peaks at promoter regions indeed showed a higher level of conservation than those not found at promoters, we found promoter and non-promoter aCNEs shared very similar phastCon scores. We also obtained similar results regarding GC content, accessibility profiles, and TF binding enrichments for both promoter and non-promoter aCNEs.

We added the following text to the manuscript:

“As aCNEs were derived from CNE datasets, in which all coding sequences had been carefully excluded^{25,26}, we found that aCNEs were depleted for exonic regions (0.78% versus 4.7%, adjusted $P = 3.95e-50$, Fisher’s exact test) and were enriched for intronic regions (25.0% versus 21.7% adjusted $P = 5.15e-07$, Fisher’s exact test) compared to all ATAC-seq peaks. We did not see an enrichment for promoter regions in aCNEs compared to all ATAC-seq peaks (9.34% versus 9.42%, adjusted $P = 1$, Fisher’s exact test). While ATAC-seq peaks at promoters were more conserved than those not at promoters ($P=7.82e-289$, Wilcoxon test), promoter aCNEs did not show higher sequence constraint than non-promoter aCNEs ($P=0.19$, Wilcoxon test) (Supplementary Fig. 3D).

“When we separated promoter and non-promoter aCNEs for analysis we observed similar results regarding interval length, GC content and open chromatin signal intensity (Supplementary Fig. 3E, F).”

Page 17

“Overall, DNA binding factor enrichment in promoter and non-promoter aCNEs were correlated ($R=0.8$, $P < 2.2e-16$), with the top enriched factors largely overlapping (Supplementary Fig. 7B-D).”

3) At many places the manuscript lacks statistical information, i.e. is the overlap with the VISTA data significant? The authors should consequently include random genomic regions with a comparable size distribution in the analysis to be able to test the significance and degree of conservation.

We have gone through the manuscript again and have performed additional appropriate statistical tests including:

Page 7

“The average of the assembly of all single-cell transcriptomes correlated well with the bulk mRNA-seq results (Pearson’s correlation $R=0.93$, $P < 2.2e-16$) (Supplementary Fig. 2A)”

Page 8

“Our ATAC-seq peaks significantly overlap with active chromatin marks found at promoters (H3K4me3, $P = 0.001$, permutation test by GAT³⁰) and enhancers (H3K27ac, $P = 0.001$; H3K4me1, $P = 0.001$, permutation test by GAT³⁰) that were previously identified from ChIP-seq experiments on whole zebrafish embryos of similar stages⁶”

“Overall both the GFP+ specific and GFP- specific open chromatin regions were depleted for proximal promoter (GFP+: adjusted $P = 1.83e-21$, GFP-: adjusted $P = 6.70e-15$, Fisher’s exact test) and exonic regions (GFP+: adjusted $P = 3.51e-17$, GFP-: adjusted $P =$

4.81e-11, Fisher's exact test) and enriched for introns (GFP+: adjusted $P = 7.35e-12$, GFP-: adjusted $P = 1.00e-10$, Fisher's exact test) as compared to the genomic distribution of all ATAC-seq peaks (Supplementary Fig. 3B)."

Page 11

"As aCNEs were derived from CNE datasets, in which all coding sequences had been carefully excluded^{25,26}, we found that aCNEs were depleted for exonic regions (0.78% versus 4.7%, adjusted $P = 3.95e-50$, Fisher's exact test) and were enriched for intronic regions (25.0% versus 21.7% adjusted $P = 5.15e-07$, Fisher's exact test) compared to all ATAC-seq peaks. We did not see an enrichment for promoter regions in aCNEs compared to all ATAC-seq peaks (9.34% versus 9.42%, adjusted $P = 1$, Fisher's exact test). While ATAC-seq peaks at promoters were more conserved than those not at promoters ($P=7.82e-289$, Wilcoxon test), promoter aCNEs did not show higher sequence constraint than non-promoter aCNEs ($P=0.19$, Wilcoxon test) (Supplementary Fig. 3D)."

"First, we found conserved ATAC-seq peaks had a wider boundary (average 706 bp compared to 475 bp, $P = 0.001$; permutation test), stronger open chromatin signals ($P = 2.84e-271$; Wilcoxon test) and slightly higher GC content (average 46% compared to 44%, $P = 0.001$; permutation test) than the bulk zebrafish ATAC-seq peaks (Supplementary Fig. 3E, F)."

Page 12

"To gain insight into the cardiac enhancer activity of aCNEs, we compared the 281 GFP+ human aCNEs to putative heart enhancers predicated using curated epigenomic data²³. We found a significant overlap between them (78/281, $P=1.68e-3$, Fisher's exact test) when using all human DNase I sites as background."

"While less than 35% of the human regions (953/2787) profiled by VISTA were functionally validated as positive enhancers, two thirds of human aCNEs (640/958) tested were reported as positive enhancers, indicating a significant enrichment of active developmental enhancers amongst aCNEs ($P = 9.02e-26$, Fisher's exact test)."

4) The heart is a heterocellular organ and the main cell types arise from a common cardiac progenitor. A large fraction of genes, TFs and gene ontology terms (GREAT) linked to conserved regions in the heart are cardiomyocyte specific. Is the conservation of enhancer elements in the heart different between cell types? The authors should reanalyze existing human and murine cardiomyocyte data in comparison to heart tissue to address this question.

This is an interesting question. In our introduction we cited work from Nord et al. 2013 who showed that enhancers in mesoderm cells, derived from embryonic stem cells, show higher evolutionary constraint than the enhancers identified after organogenesis. To directly address this comment we performed the following analysis:

DHSs identified in human cardiomyocytes (wgEncodeEH000519) and human cardiac fibroblasts (wgEncodeEH000513) were downloaded from UCSC (http://hgs.v.washington.edu/cgi-bin/hgTrackUi?hgsid=2731596_MXj1v22xjvU6xzC53Vy6MJEAbVDy&c=chr7&g=wgEncodeAvgDnaseUniform). These peaks were uniformly processed through the ENCODE pipeline thus served as good datasets for comparison. Using the same pipeline for aCNE identification, we found that 866 out of 193,489 DHSs in HCM (0.45%) and 795 out of 174,781 DHS in HCF (0.45%) were aCNEs conserved with zebrafish. There was no significant difference between the percentage of aCNEs identified in these two different cardiac cell types ($P=0.749$, Fisher's exact test).

5) Some sentences are not logic. For example: Of these 6294 ATAC-seq peaks 176 were GFP+.

We have corrected this and fixed other grammatical issues in the text.

6) Please use a different type of Venn diagram (Fig. S3a), which allows to identify the individual intersections more easily.

We appreciate and fully agree with this comment. We have now replaced the Venn diagram in Supplementary Fig. 3A with an UpSet plot, which is much more interpretable.

Reviewer #3 (Remarks to the Author):

In the manuscript entitled “Heart enhancers with deeply conserved regulatory activity are established early in development” Yuan and colleagues perform comparative epigenomic analysis to identify evolutionarily conserved cardiac enhancer. They introduced the mouse smarcd3-F6 enhancer, the enhancer of early specified cardiac precursors, into zebrafish and established transgenic animal for early cardiac precursor specific RNA-seq and ATAC-seq experiments. Exploiting an impressive amount of transgenic tools, they identify multiple cardiac enhancers, some of which are well-conserved in both zebrafish and human and show activity in zebrafish heart. The authors also analyzed usages of a large number of conserved accessible chromatin (aCNEs) in 127 human tissue/cell types.

Overall, this is an exciting study to understand the function of the conserved sequences between fish and mammal. Introducing mammalian enhancer into zebrafish and determining conserved enhancers across species is a conceptually well-designed approach. Their bioinformatics analysis can apply to other projects and can provide a new direction to identify novel enhancers. The total number of established transgenic lines are truly impressive, and these lines can be valuable

resources to elucidate gene regulatory networks underlying early cardiac development.

Despite these strengths, there are several areas in which this manuscript could be significantly improved.

We thank the reviewer for appreciating our work and for the constructive comments.

Major Comments

1. Enhancer activity of the *smarcd3-F6:EGFP* transgenic line

What is native EGFP expression without using immunostaining or in situ hybridization (ISH)? The authors FACS-sorted with EGFP signal, suggesting that EGFP can be observed without further staining. The authors should provide a figure of native EGFP expression to demonstrate which cells are sorted for ATAC-seq and RNA-seq. In addition, the authors need to briefly mention why they detected enhancer activity by ISH and immunostaining rather than live EGFP imaging.

We now show a native GFP image of a 10 hpf *Tg (Smarcd3-F6:EGFP)* embryo in revised Fig. 1.

The reasons that we used ISH and immunostaining instead of imaging native GFP:

- GFP protein takes hours to be translated, folded properly, accumulate and eventually become fluorescent, while early zebrafish embryonic development progresses very rapidly. When the enhancer just became activated (early-mid gastrulation stage), the GFP signal was not strong enough to provide clear fluorescent images that reflect the real-time activity of the enhancer. ISH serves as a sensitive and robust approach to detect transgene expression.
- We used immunostaining to compare *Tg (Smarcd3-F6:EGFP)* and *Tg (nkx2.5:ZsYellow)* expression as the native *ZsYellow* fluorescent signal was not bright enough to be detected at 13 hpf.

We added a brief description of why we used ISH to detect *gfp* to the manuscript (page 6):

“Due to the lag time between transcription and GFP accumulation, we conducted RNA *in-situ* hybridization against *gfp* in order to detect enhancer activity at early developmental times. We found that *gfp* signal could be detected as early as 6 hours post-fertilization (hpf) along the embryonic margin (Fig. 1B)”

2. Figure 2. D. The authors performed a known-down experiment of *gata5/6* to examine whether *smarcd3-F6* enhancer activity requires the function of *Gata5* and *Gata6*. However, they should perform the following control experiments. 1) Known-

down of sox gene(s) and then examine EGFP expression: This experiment will confirm that disruption of sox gene, binding motifs of which are enriched in ATAC-seq profiles of GFP negative cell, would not affect the activity of smarcd3-F6 enhancer. 2) Expression of ectoderm gene(s) in *gata5/6* KD embryos. *emilin3a*, *nkx2.9*, or *lhx2b* would be used as an ectoderm gene.

A) We have done these additional experiments. We chose to knock down Sox32, an early activated transcription factor whose morpholino has been validated and widely used in previous studies (1-4). We repeated our *Gata5/6* knockdown together with the control Sox32 knockdown. Unlike *Gata5/6* knockdown, Sox32 knockdown did not reduce the GFP signals in *Tg (Smarcd3-F6: EGFP)* line. These new results are now included in Figure 2D and described in the text on page 9 as:

“In contrast, knocking down an early active SOX transcription factor (Sox32)⁴³ did not ablate GFP signal (Fig. 2D).”

B) We also examined the expression of *emilin3a* and *nkx2.9* in *gata5/6* KD embryos at 10 hpf (see data below). As expected, we did not observe significant changes of their expression upon *gata5/6* KD. After conducting the ISH, we realized some of the GFP-genes in Fig. 1F were not properly labeled. *emilin3a*, *noto* and *pmp22b* were primarily expressed in notochord (part of axial mesoderm) instead of neural tubes. So we revised the Fig. 1F accordingly. Although this did not change our conclusion, we appreciated reviewer for helping us identify this oversight.

- (1) Dickmeis, T. *et al.* A crucial component of the endoderm formation pathway, CASANOVA, is encoded by a novel sox-related gene. *Genes Dev.* **15**, 1487–92 (2001).
- (2) Ye, D., Xie, H., Hu, B. & Lin, F. Endoderm convergence controls subduction of the myocardial precursors during heart-tube formation. *Development* **142**, 2928–2940 (2015).
- (3) Paksa, A. *et al.* Repulsive cues combined with physical barriers and cell–cell adhesion determine progenitor cell positioning during organogenesis. *Nat. Commun.* **7**, 11288 (2016).
- (4) Chou, C.-W., Hsu, H.-C., You, M., Lin, J. & Liu, Y.-W. The endoderm indirectly influences morphogenetic movements of the zebrafish head kidney through the posterior cardinal vein and VegfC. *Sci. Rep.* **6**, 30677 (2016).

3. ZaCNE and HaCNE transgenic lines

Transgenic zebrafish lines generated with ZaCNEs and HaCNEs are valuable reagents to elucidate novel sub-population of cardiac progenitor cells and related cardiac TF motif characterization in cardiac enhancers. For improving their analysis, the author should perform the following experiments:

1) Expression patterns in early time point

In Discussion, Page 18, line 9, the authors described that they tested enhancer activity in stable zebrafish lines at 24 hpf. However, this data is missing. All EGFP expressions in figure 3D, 4A, 4B, Sup. Fig. 5, and Sup. Fig. 6 are taken at 48 hpf, not 24 hpf. The authors should provide 24 hpf images, which are more valuable than 48 hpf. 48 hpf does not represent the early developmental stage (the zebrafish phylotypic stage), which this paper addresses. All enhancers are identified using 10 hpf cardiac progenitor cells, so I assume that EGFP expression can be detected by 24 hpf without additional staining.

We thank the reviewer for drawing our attention to the lack of 24 hpf data, which clearly is most relevant to the phylotypic period. To address this comment systematically, we decided to re-cross the 18 stable transgenic lines that we generated and capture their expression pattern at 24 hpf. The results have been included in Supplementary Fig. 5E.

We present the results on page 14:

“We also assayed the presumptive phylotypic period (24 hpf) and found that 16/18 drove expression in the linear heart tube (Supplementary Fig. 5E).”

We kept our analysis and categorization based on 48 hpf data since at 48 hpf, the major chamber structure of a zebrafish heart is established. This allowed us so to closely examine and broadly categorize which anatomical regions of heart were labeled by the

aCNEs. Images at 24 hpf are challenging to categorize in this way since only the linear heart tube forms at this time point, with no clear boundaries of the future compartments.

2) I am wondering why the authors again use ISH to confirm enhancer activity at the earlier time point such as 24 hpf. They need to briefly mention why they used ISH to avoid the reader's confusion that ISH is likely a common method to detect enhancer activity with EGFP tg lines.

We updated our manuscript with brief explanation of why we used ISH to characterize enhancer activity.

On page 6 we added:

“Due to the lag time between transcription and GFP accumulation, we conducted RNA *in-situ* hybridization against *gfp* in order to detect enhancer activity at early developmental times. We found that *gfp* signal could be detected as early as 6 hours post-fertilization (hpf) along the embryonic margin (Fig. 1B)”

As the perdurance of GFP protein in zebrafish embryo tend to be longer than the developmental time window (2 days) that we are interested, we used RNA ISH to get a higher temporal resolution of the enhancer activity. Due to the quick turnover of RNA transcript, results from RNA ISH represent a relatively “real-time” readout of the enhancer activity.

On page 15 we added: “Due to the long perdurance of GFP protein, we next used *gfp* RNA *in-situ* to characterize the spatiotemporal dynamics of the aCNE activity.”

3) Activity of HaCNEs: The authors generate stable transgenic lines with 4 HaCNEs, but their expression patterns are not categorized. The authors need to provide what kind of category enhancers they are. If HaCNEs exhibit different EGFP expression patterns compared to their corresponding zebrafish enhancers, it would be interesting since analyzing sequence differences between ZaCNE and HaCNE can reveal the important cardiac motifs.

This is an interesting question which is also related to the question asked in comment 4 below. We have now categorized all the HaCNEs expressed in fish.
(page 15)

“At 48 hpf, the GFP expression in the four human aCNE transgenic lines was seen in both chambers and would be classified as pan-cardiac enhancers (category I), but we noticed that the human sequences tended to drive weaker or more mosaic expression than their zebrafish orthologous sequences (Fig. 4A, B and Supplementary Fig. 6A, B).”

For this reason, combined with a limited number of examples were not able to make obvious sense of the motif differences. However we had more success with the related question below.

4) The authors described that they found 4 major categories with 17 lines. Is there any characteristic motif composition representing each category? It would be good to present maps highlighted cardiac TF motifs for at least 8 CNE (ZaCNE1, ZaCNE5, ZaCNE19, ZaCNE20, HaCNE1, HaCNE5, HaCNE19, and HaCNE20). Where are GATA4, TBX5, NKX2.5, and/or HAND2 motifs in each enhancer? How far are these motifs apart from each other in both zebrafish and human enhancers? How many copies of each motif? This analysis may elucidate novel signatures of cardiac progenitor sub-population.

This is an interesting question that we overlooked. Although we have a limited number of examples in each category, we were able to find significantly enriched motifs in three categories. The results of this analysis can now be found in Supplementary Fig. 6B.

Page 14-15:

“We asked if certain motifs were enriched in each enhancer category. With the exception of category III (atrioventricular canal), which only contains two sequences, enriched motifs were identified in all other categories. GATA, SMAD, RAR/RXR, ZNF263 and RREB1 motifs were found to be enriched in more than one enhancer category, suggesting they may represent shared features of early cardiac enhancers (Supplementary Fig. 5B). Category IV (outflow tract enhancers) had the most distinct motif signature, with FOX (FOXO1, FOXP1, FO XK1 etc.) and homeobox (HOXA2, EMX2, PDX1, etc.) motifs showing strong enrichment (Supplementary Fig. 5B). Interestingly, Foxp1 has been shown to be required for outflow tract morphogenesis⁵⁶”

4. Figure legend 5

Current Figure Legend 5 is not related. The authors must change it. For instance, (A) does not indicate Heatmap. It is about GREAT enrichment analysis.

We thank your for noticing this. This major oversight on our part has been corrected.

Minor comments:

1. Supplementary figure 1, Figure legend (A)

Superscript 1 and 2 would indicate references, but I couldn't find it. I believe that the authors will provide references used for supplementary figures in the final version. In addition, I could not find how the authors get the H3K27ac data of MES, CP, and CM. In the Method of “Processing and analyzing of published mouse cardiac TF ChIP-seq data”, there is no description of these profiles. In addition, Method does not include any H3K27ac ChIP-seq analysis.

The reference bibliography has been added to the supplementary figures at the end. Processing of H3K27ac ChIP-seq data has been added to the Method section.

Page 41:

“To generate tracks shown in Supplementary Fig. 1A, H3K27ac¹⁹ and cardiac transcription factor (GATA4, NKX2.5, TBX5)²² ChIP-seq data were used. Replicates for each factor each condition were merged by running Samtools¹¹⁴ (merge, Version 1.2) on the alignment files (in bam format) provided by the authors. Then BEDtools¹¹³ (genomecov, Version 2.19.1) were used to compute the reads coverage based on the merged bam files for track display.”

2. Figure 1 and page 6 line 3

To eliminate the possible effect of the maternally derived mRNA, in situ hybridization image of very early time point embryos are required (earlier than 6 hpf).

Thank the reviewer for pointing this out. In fact, the *Smarcd3*-F6 enhancer is active maternally in the zygote (embryos from female *Tg (Smarcd3-F6: EGFP)* crossed to WT male have GFP fluorescence at one-cell stage). To eliminate this maternal effect, all experiments that we have done used embryos from male *Tg (Smarcd3-F6: EGFP)* to WT female cross. We clarified this important point in the Method section.

Page 23-24:

“Since we noticed maternal expression of GFP in the *Tg (Smarcd3-F6: EGFP)* line, we only used male *Tg (Smarcd3-F6: EGFP)* fish crossed to female WT fish for our experiments.”

We have also done *gfp* ISH at earlier time points than 6 hpf. We did not see *gfp* transcripts even at 5.5 hpf (50% epiboly to germ ring stage). So, we concluded in the paper that “... signal could be detected as early as 6 hours post-fertilization (hpf) along the embryonic margin” (Page 6)

3. Page 6 line 8 – 9 “Co-immunostaining comparing Tg(Smarcd3-F6:EGFP) and Tg(nkx2.5:ZsYellow) expression indicated that the Smarcd3-F6 enhancer marked all cardiac mesoderm expressing nkx2.5 at early somite stages (13 hpf) (Fig. 1C).” Although most nkx2.5:ZsYellow positive cells are labeled by EGFP, there are some ZsYellow positive and smarcd3-F6:EGFP negative cells. Thus, “all cardiac mesoderm” should be toned down (for example “almost”).

We agree and have toned this down accordingly: “.... *Smarcd3*-F6 enhancer marked almost all cardiac mesoderm....”

4. Page 9 line 10

At least one base pair -> it could be fifteen.

We apologize for the confusion. To clarify, we removed “by at least one base pair” in the text and now say “We associated zebrafish ATAC-seq peaks to CNEs if they overlapped a zebrafish-human or zebrafish-mouse CNE.”

We assumed that the impression of 15 bp came from page 34 “zCNEs that overlaps a zebrafish-human well-aligning window²⁵ by at least 15bp were defined as direct aligned.” This is the criteria Hiller et al. used for defining zCNEs.

5. Page 12, Second paragraph, line 7, 10

Experimentally determined binding sites:

There are no descriptions what these binding sites are. Only referring references are not enough. The authors should mention the exact sequences of “experimentally determined binding sites” in the supplementary data.

We have included a table of the all public data sets we used, their accession numbers and whether or not we re-processed the data ourselves (see Method). We have now included the peaks from all reprocessed data in Supplementary Table 4.

6. Figure 3D and supplementary figure 5

It would be good to make another column for category (expression pattern) in supplementary table 3.

Done, now included in Supplementary Table 5

7. Page 15, line 16: Fig 3D -> Fig. 5B

Done

8. Page 16, line 4: the the -> remove one the

Done

9. Sup. Fig. 4, 5: There is no annotation about time point of embryos they used for imaging. I assume 48 hpf. The authors should annotate the time point.

Done

10. Method: No data about antibodies.

This information is now included on page 25 ((α -RCFP 1:500, Clontech Cat# 632475; α -GFP 1:1000, ThermoFisher Cat# A-11120))

Reviewer #1 (Remarks to the Author):

As I stated before, this is an important research report, thoroughly investigated and clearly reported. The minor issues I raised have been fixed.

Reviewer #2 (Remarks to the Author):

I thank the authors for providing the revised version of the manuscript. The review process has really improved the manuscript.

- 1) The authors have added the requested statistical info.
- 2) The authors did not add the requested peak tracks to figure 2A. Adding a supplementary table of all detected peaks is nice, but does not help the reader to judge the results and peak calling algorithms. It would be really helpful to add these tracks to Fig. 2A.
- 3) For the revised version of the manuscript the authors added the requested separate analysis of promoter and distal regions. This analysis is of great value for the conclusions of the manuscript. Unfortunately, the authors do not provide an experimental identification of active regulatory regions (i.e. ChIP-seq) to clearly discriminate enhancers from inactive open chromatin. Remarkably, the authors state in the response to the reviewer's questions, that the paper focuses on accessible conserved non-coding elements and the authors see value in including all such regions in their analysis. Maybe the authors should state this point as a study limitation in the main manuscript.

Reviewer #3 (Remarks to the Author):

I appreciate the authors for performing all experiments to resolve my comments. I don't have any more comments. This manuscript is nicely formulated, and I highly recommend for publication of Nature Communication. I expect it will provide many insights into the regulatory elements controlling cardiac development.

REVIEWERS' COMMENTS:

Reviewer #1 (Remarks to the Author):

As I stated before, this is an important research report, thoroughly investigated and clearly reported. The minor issues I raised have been fixed.

Reviewer #2 (Remarks to the Author):

I thank the authors for providing the revised version of the manuscript. The review process has really improved the manuscript.

- 1) The authors have added the requested statistical info.
- 2) The authors did not add the requested peak tracks to figure 2A. Adding a supplementary table of all detected

peaks is nice, but does not help the reader to judge the results and peak calling algorithms. It would be really helpful to add these tracks to Fig. 2A.

These have now been added.

3) For the revised version of the manuscript the authors added the requested separate analysis of promoter and distal regions. This analysis is of great value for the conclusions of the manuscript. Unfortunately, the authors do not provide an experimental identification of active regulatory regions (i.e. ChIP-seq) to clearly discriminate enhancers from inactive open chromatin. Remarkably, the authors state in the response to the reviewer's questions, that the paper focuses on accessible conserved non-coding elements and the authors see value in including all such regions in their analysis. Maybe the authors should state this point as a study limitation in the main manuscript.

We have one such a comment at the end of the discussion:

"While more work remains to be done to dissect the in vivo, spatial-temporal expression patterns and function of anciently conserved vertebrate enhancers, regions of deeply conserved open chromatin represent a solid foundation from which the regulation and evolution of the vertebrate body plan can be explored."

We have now added a second, more specific, statement in the discussion:

"Further experiments, such as ChIP-seq for individual transcription factors and posttranslational histone modifications will help understand the spatial and temporal dynamics activity of these aCNEs."

Reviewer #3 (Remarks to the Author):

I appreciate the authors for performing all experiments to resolve my comments. I don't have any more comments. This manuscript is nicely formulated, and I highly recommend for publication of Nature Communication. I expect it will provide many insights into the regulatory elements controlling cardiac development.